# A computational model of liver tissue damage and repair

**Priyom Adhyapok**[1,2]*, **Xiao Fu**[3], **James P. Sluka**[1,4], **Sherry G. Clendenon**[1,4], **Victoria D. Sluka**[1¤a], **Zemin Wang**[5¤b], **Kenneth Dunn**[6], **James E. Klaunig**[5], **James A. Glazier**[1,4]

**1** Biocomplexity Institute, Indiana University, Bloomington, IN, United States of America, **2** Department of Physics, Indiana University, Bloomington, IN, United States of America, **3** The Francis Crick Institute, London, United Kingdom, **4** Department of Intelligent Systems Engineering, Indiana University, Bloomington, IN, United States of America, **5** School of Public Health, Indiana University, Bloomington, IN, United States of America, **6** School of Medicine, Indiana University, Indianapolis, IN, United States of America

¤a Current address: Institute of Archaeology, University College London, London, United Kingdom
¤b Current address: Food and Drug Administration, Washington D.C., United States of America
* priyom.adhyapok@gmail.com

**Data Availability Statement:** All relevant data are within the manuscript and its Supporting Information files.

**Funding:** This work was supported by National Institute of Health (www.nih.gov) grants; NIH U01

## Abstract

Drug induced liver injury (DILI) and cell death can result from oxidative stress in hepatocytes. An initial pattern of centrilobular damage in the APAP model of DILI is amplified by communication from stressed cells and immune system activation. While hepatocyte proliferation counters cell loss, high doses are still lethal to the tissue. To understand the progression of disease from the initial damage to tissue recovery or death, we computationally model the competing biological processes of hepatocyte proliferation, necrosis and injury propagation. We parametrize timescales of proliferation ($\alpha$), conversion of healthy to stressed cells ($\beta$) and further sensitization of stressed cells towards necrotic pathways ($\gamma$) and model them on a Cellular Automaton (CA) based grid of lattice sites. 1D simulations show that a small $\alpha/\beta$ (fast proliferation), combined with a large $\gamma/\beta$ (slow death) have the lowest probabilities of tissue survival. At large $\alpha/\beta$, tissue fate can be described by a critical $\gamma/\beta*$ ratio alone; this value is dependent on the initial amount of damage and proportional to the tissue size $N$. Additionally, the 1D model predicts a minimum healthy population size below which damage is irreversible. Finally, we compare 1D and 2D phase spaces and discuss outcomes of bistability where either survival or death is possible, and of coexistence where simulated tissue never completely recovers or dies but persists as a mixture of healthy, stressed and necrotic cells. In conclusion, our model sheds light on the evolution of tissue damage or recovery and predicts potential for divergent fates given different rates of proliferation, necrosis, and injury propagation.

## Introduction

The liver performs a variety of essential functions including the clearance and metabolism of toxins in the bloodstream. It is generally considered to consist of repeated units called lobules, with a human liver containing a billion lobules. Individual lobules are "plumbed" in parallel

GM111243 Development of a Multiscale
Mechanistic Simulation of Acetaminophen Induced
Liver Damage funded JPS, SGC, VDS, ZW, KD,
JEK, JAG; NIH R01 GM122424 Competitive
Renewal of Development, Improvement and
Extension of the Tissue Simulation Environment -
CompuCell3D funded JPS, JAG; and NIH U24
EB028887 Dissemination of libRoadRunner and
CompuCell3D funded JPS, JAG. The funders had
no role in study design, data collection and
analysis, decision to publish, or preparation of the
manuscript.

**Competing interests:** The authors have declared
that no competing interests exist.

and a portion of blood passing through the liver passes through a single lobule before exiting. Individual lobules are supplied with both arterial blood (form the heart and lungs) and venous blood (from the stomach and intestines). These input blood supplies occur at the periphery of the lobule in regions called portal triads (PTs). Blood flows from the PT in the periportal region towards the pericentral region and drains into a central vein (CV). The lobule parenchyma consists of a dense network of liver capillary vessels, called sinusoids, that carry the blood from the PTs to the CV. In general, individual hepatocytes, the major parenchymal cells of the liver, are in contact with two or more sinusoids.

The liver displays a range of patterns of damage depending on the causative agent(s). Damage from drugs (drug induced liver damage, DILI), infections such as hepatitis, and damage due to environment and lifestyle (obesity, alcoholic cirrhosis, pollutants) show a range of damage patterns and progression timelines. Damage patterns may be an emergent property of the spatially varying expression levels of hepatocyte metabolic process (such as cytochrome P450 expression), a result of difference in microdosimetry (dose variation over the scale of several cells) [1,2], or emergent from the spatial organization of a liver lobule.

A well studied cause of liver damage is Acetaminophen (APAP, paracetamol) overdose. APAP is one of the most widely used analgesic and antipyretic drugs globally [3]. While APAP is safe when the therapeutic dosage (4 g/day) is not exceeded [4], overdose can cause dose-dependent hepatocellular necrosis [5,6]. APAP toxicity underlies over 40% of all acute liver failure (ALF) in the United States [7], the United Kingdom and Europe [8] and is responsible for more ALF than all other etiologies combined [9].

## APAP induced liver injury

The majority of a therapeutic dose of APAP is metabolized by glucuronidation or sulfation and rapidly excreted in urine. The remainder is converted by CYP2E1 to the reactive compound N-acetyl-p-benzoquinone imine (NAPQI) that readily reacts with sulfhydryl groups on proteins. At therapeutic APAP dosages the small amount of NAPQI produced is detoxified by binding to the cysteine thiol of glutathione (GSH) and excreted from the liver via bile. With higher doses of APAP the glucuronidation and sulfation pathways become saturated and a larger proportion of the APAP is metabolized by CYP2E1 to NAPQI. Cellular GSH then becomes depleted and the excess NAPQI is free to form harmful adducts with other proteins within hepatocytes [5,10]. The most harmful effects result from NAPQI binding to mitochondrial proteins [11–13].

Intracellular events due to APAP induced hepatotoxicity, as outlined by Jaeschke et al [14] can be characterized by a phase of initiation in which APAP overdose results in NAPQI formation and GSH depletion [5,15,16], followed by APAP protein adduct formation, mitochondrial stress and compromised cellular respiration [10]. This is followed by the amplification of the injury in which oxidant stress results in mitochondrial DNA damage and loss of mitochondrial membrane integrity, collapsing the membrane potential and disabling ATP production. Without cellular energy production, membrane integrity is lost and nuclear DNA fragments, resulting in necrotic cell death.

The initial APAP injury is pericentral and coincides with the pericentral localization of high CYP2E1 expression. This damage is characterized by the early appearance of stressed cells near the central vein (CV) [15,17]. At a sufficiently high APAP dose, the injured region expands towards the periportal region [18] and can result in complete liver failure.

Some researchers have attributed this progression of damage to intercellular communication mediated by gap junctions. Experiments with mice deficient in connexin 32, a key gap junction protein, given a normally toxic dose of APAP, showed reduced necrosis and that

necrosis did not propagate throughout the lobule [19]. Gap junctions appear to amplify the damage by propagating oxidative stress signals between adjacent hepatocytes. Other experiments have also seen synchronized deaths in hepatocytes mediated by gap junctions [20].

In response to APAP induced injury, hepatocytes attempt to compensate for the loss in liver mass by proliferating [21]. In general, this regeneration is related to cytokines produced by the innate immune system of the liver.

Immune system components are activated as necrotic hepatocytes release components, including nuclear DNA fragments, formyl peptides and HMGB1 (High-Mobility Group Box-1 Protein), that act as damage-associated molecular patterns (DAMPs) [14,22]. HMGB1 proteins bind to the liver resident macrophages, called Kupffer cells, through toll-like receptors (TLRS) [23], inducing further cytokine secretion that recruits neutrophils and Ly6C high monocytes. These monocytes and Kupffers secrete pro-inflammatory cytokines such as TNF-$\alpha$, and IL-6 that also have pro-mitotic effects by priming healthy hepatocytes to be more responsive to growth factors [21,24–27]. At the same time these cytokines also activate death pathways in cells [28,29]. The determining factor of whether a hepatocyte proliferates or undergo necrosis or apoptosis may depend on the amount of intracellular ATP [30].

These immune system components have the potential to release additional reactive oxygen species causing additional damage [31]. Similar behavior can be attributed to neutrophils that come in to clear the dead cells, and are also capable of releasing a variety of oxidants [32,33]. While neutrophils don't normally damage healthy cells they can make mistakes and kill normal and stressed cells in the vicinity of damage [33,34]. Thus, although the sterile inflammatory response is necessary to remove cellular debris and activate liver regeneration, this response also has the potential to aggravate the injury.

These observations raise the question; what tips the balance where the same set of cell behaviors that are needed for tissue repair and survival can, in some cases, lead to widespread cell death and irreparable tissue damage? To develop an understanding of the progression of liver damage in this system we have developed a Cellular Automata (CA) model of hepatocyte injury propagation, death and proliferation using a one dimensional (1D) linear chain and two dimensional (2D) hexagonal grids of simulated hepatocytes with key parameters associated to the timescales of these three processes. CA based models have been previously used to model a wide variety of processes including stock market dynamics [35], spread of forest fires [36,37], cancer progression [38], proliferation in migratory cells [39], invasion [40], neurosphere growth [41]. These models have been shown to be capable of producing rich parameter sets of outcomes and have been additionally used to describe phase transitions of infection dynamics [42,43] and traffic flow [44].

## Approach

Our CA model consists of hepatocytes as discrete sites on either a one dimensional (1D) chain of cells, or as a two dimensional (2D) grid on a hexagonal lattice. Both 1D and 2D simulations represent typical representations of a liver lobule.

Our 1D model consists of a string of hexagonal hepatocytes representing a simple hepatocyte cord running from a portal triad to the central vein (PT to CV). A 2D simulation consists of a regular hexagonal array of cells representing a hexagonal lobule, centered on a CV, consisting of hexagonal hepatocytes. The 1D and 2D arrangements allow us to explore the spatial effects on the different processes based on the positional inputs and number of neighbor cells.

A cell lattice site can be a cell in a healthy (H) state, a cell in a stressed (S) state, or an empty site where a cell has died (D). Similar to what has been implemented in [45], a healthy cell can be converted to a stressed one due to the presence of other stressed cells in the neighborhood.

A stressed cell eventually dies, leaving behind an empty space. However, recent intravital observations have also revealed that stressed hepatocytes can recover from APAP induced injury through mitochondrial repolarization [16]. Our current model does not include this process but it could be included as a transition from a stressed cell back to a normal cell. Healthy cells can proliferate and repopulate neighboring empty sites, which couples cell division to the death of a neighboring cell.

Our model has three different timescales — (A) $\alpha$, associated with the cell proliferation timescale, (B) $\beta$, associated with the conversion of a healthy cell to a stressed cell, and (C) $\gamma$, associated with the process of the stressed cell dying. Varying these parameters allows us to observe the phase space of outcomes leading to either tissue recovery or complete tissue death. We measure the average number of healthy cell states as the characteristic metric at the end of each simulation.

The CA model uses stochastic transitions based on our parameters of $\alpha$, $\beta$, and $\gamma$ and the state transition rules. We initialize the CA (Fig 1A) to represent the experimentally observed first appearance of pericentral stressed cells at around 2 hours (see Fig 2H) and assume an initial region, defined by a distance from the CV, consisting completely of stressed cells. NAPQI protein adduct studies at different doses of APAP have resulted in different centrilobular

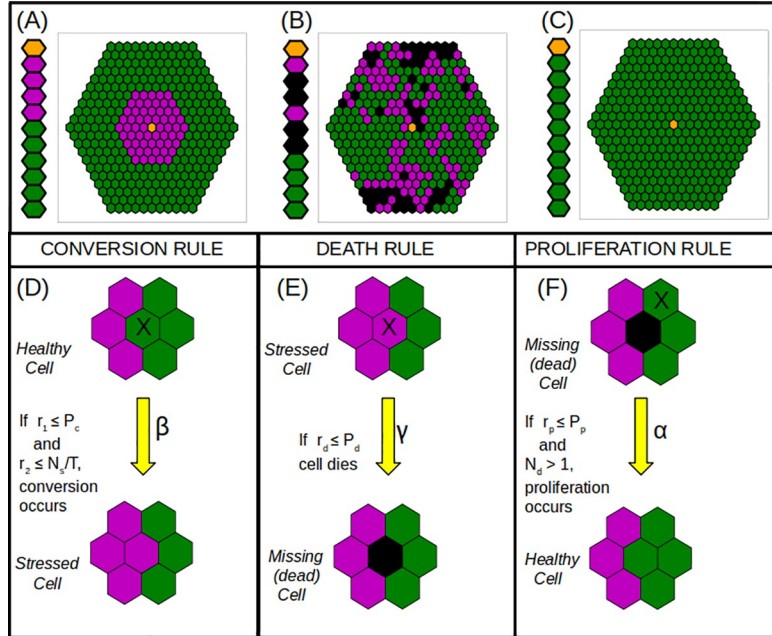

**Fig 1. Cellular automata based rules of proliferation, death and conversion.** Healthy cells are colored in green, stressed cells in magenta, empty spaces are in black, unmodeled CV is colored in orange. (A) Initial conditions for a 1D sinusoidal cord or a 2D hexagonal lobule. (B) Output of the CA model at intermediate time, at any point in time total number of sites $N$ is made of healthy ($H$), stressed ($S$) and empty spaces ($D$) left by dead cells. (C) Output of CA model of tissue survival with $H = N$, $S = 0$, $D = 0$. (D-F) CA rules of updating the model with $\alpha$ as the proliferation timescale, $\beta$ as the timescale of conversion of a healthy to a stressed cell and $\gamma$ as the timescale of stressed cell death. The cell to which the rule is being applied is marked by an X. The Conversion Rule converts a cell from normal to stressed and is dependent on the number of stressed neighbors around the healthy cell. Two random numbers are chosen - $r_1$ has to be lesser than the conversion parameter and $r_2$ has to be lesser than the total number of stressed neighbors ($N_s$) normalized over the total number of the cells' neighbors ($T$). The Death Rule kills a stressed cell if a random number is less than the death parameter $P_d$. The Proliferation Rule fills in an empty location (created by a dead cell) and will only occur if the space is next to a healthy cell. For each healthy cell, a random number $r_p$ is chosen and proliferation occurs at that time step if it is less than the proliferation parameter $P_p$ and one of its six immediate neighbors contains an empty site ($N_d > 1$).

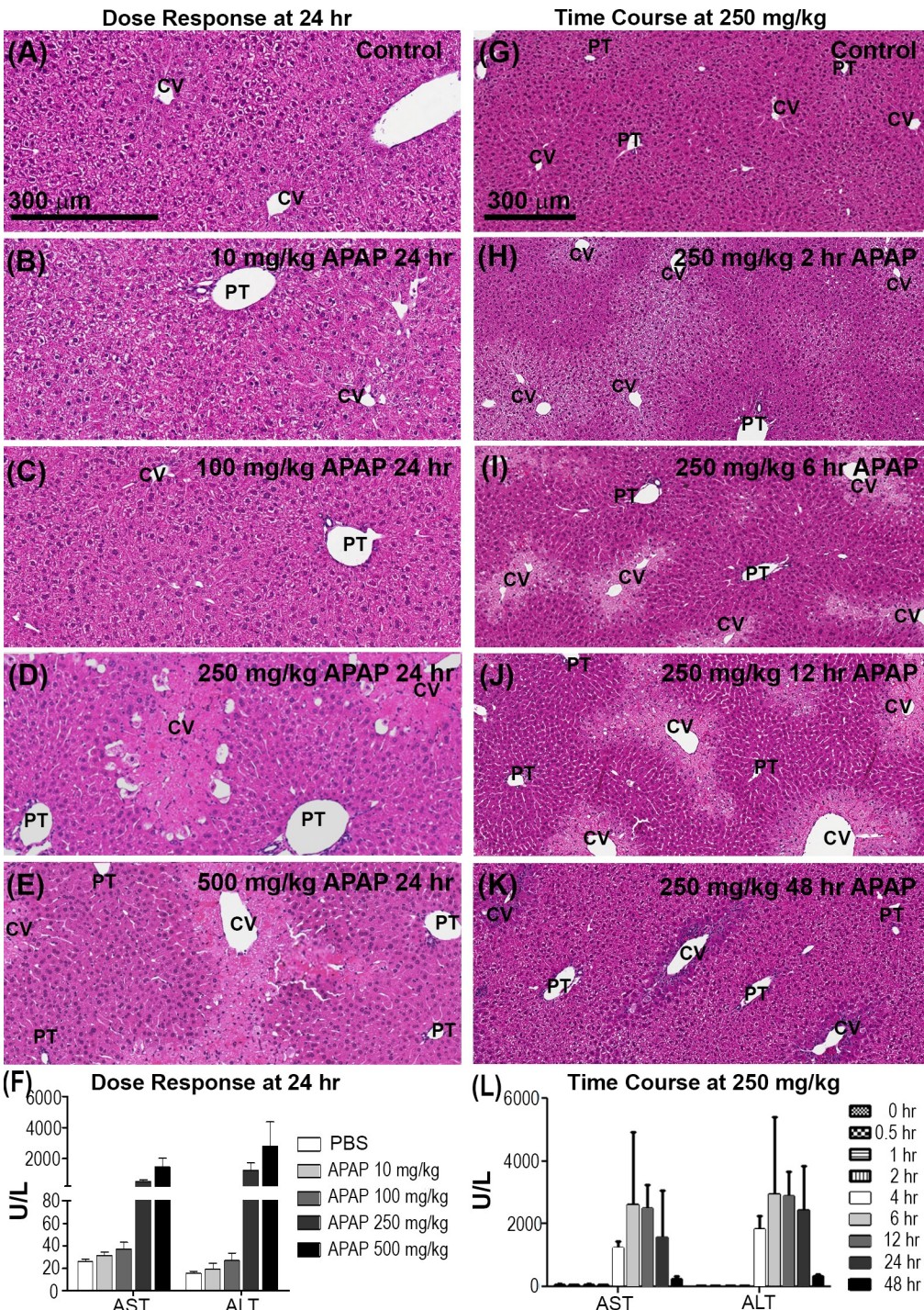

**Fig 2. Dose response, pattern and timing of hepatic necrosis and recovery after APAP.** Dose Response: (A-E) H&E staining of mouse liver 24 hours after APAP at 0, 10, 100, 250 and 500 mg/kg. Bar = 300 $\mu$m. (F) Serum AST and ALT 24 hours after APAP at 0, 10, 100, 250 and 500 mg/kg (n = 3 animals in each group). No necrosis is evident at 10, or 100 mg/kg APAP although modest increases in AST/ALT were observed. Pericentral necrosis, visible as regions of hepatocytes with lighter pink cytoplasm and condensation and loss of hepatic nuclei, has developed at 250 and 500 mg/kg and is accompanied by major increases in AST/ALT (mean ± s.e.). Time Course: (G-J, D, K) H&E staining of mouse liver at 0, 2, 6, 12, 24 and 48 hours after APAP at 250 mg/kg. (G-K) Bar = 300 $\mu$m. (L) Serum AST and ALT at 0, 2, 6, 12 and 48 hours after APAP at 250 mg/kg (n = 3 animals in each group). Data re-visualized as in Dunn et al [16]. At 250 mg/kg the cytoplasm of cells around the CVs begins to appear lighter by H&E staining as early as 30 minutes after administration of APAP. The lightened region becomes easily discernible around 2 hours after APAP treatment,

but hepatic nuclei remain intact and AST/ALT is unchanged from control. Pericentral necrosis has begun by 4 hours after APAP treatment and is accompanied by increased AST/ALT. Pericentral necrosis is prominent at 6, 12 and 24 hours and is accompanied by further increased AST/ALT. (K) By 48 hours pericentral region shows signs of recovery.

patterns by 2 hours, with higher doses showing the adduct levels across a larger area [18]. We implement this dose effect as a shift in the initial position of the boundary between healthy and stressed (or dead) cells towards the portal region, resulting in an increasing number of stressed cells and decreasing number of healthy cells.

## Rules for updating the CA

At any point in time there are $N$ total sites made up of healthy cells, stressed cells and dead/empty sites (Fig 1B). For each cell we keep track of how many of its neighbors sites are stressed ($N_S$), healthy ($N_H$), or empty (dead) ($N_D$). The automaton evolves through the interaction of these states through equally spaced time points with a time step of $\Delta t$. Each of these cells can choose to update their state according to the three rules described below at every time point.

**(A) Proliferation.**   Hepatocytes in the liver try to proliferate to maintain a constant mass. In our model only healthy cells can proliferate. We assume a characteristic time scale of $\alpha$ associated with the proliferation process, such that the probability of a hepatocyte dividing in a computation time step becomes $P_p = \Delta t/\alpha$. Since this is a probability, we require that $0 < \Delta t \leq \alpha$. All the other parameters associated with timescales will also obey this relation.

Additionally, we assume that this process is limited by space and that the ability of a cell to divide is dependent on its immediate cell density. This is based on the observation that division occurred adjacent to necrotic regions and similar assumptions have been used in [40,41], based on cells' ability to mechanically sense its neighborhood [46]. Contact-inhibited growth has also been seen in cultured hepatocytes in [47].

Computationally to achieve both of these criteria, we begin by picking a random number $r_p$ from a uniform distribution for each healthy hepatocyte and checking if $r_p \leq P_p$. If yes, the healthy cell is added to a queue. Next, for all the cells in the proliferation queue we check if each hepatocyte has free space to move into by checking for the presence of empty spaces left behind by dead cells. In the 1D case, each healthy hepatocyte can look at its two nearest neighbors (to its left and right) while in the 2D case hepatocytes can look at six of its nearest neighbors. We make a binary choice here, if any of its neighboring sites are empty the cell will choose to divide and if not, the cell won't. Upon division the hepatocyte will take the place of the empty site. If multiple cells pick the same empty site for division, the cells are randomly shuffled and one cell is picked.

**(B) Cell death.**   Stressed cells undergo necrosis by losing plasma membrane integrity. We assume that there is a characteristic time scale associated with this process of stressed cells dying, denoted by $\gamma$. We assume that this process is independent of neighbor density and any stressed cell can die with a probability of $P_d = \Delta t/\gamma$. Similar to choosing cells for proliferation, we pick a random number $r_d$ for each dead cell and check if $r_d \leq P_d$.

**(C) Conversion of a healthy to a stressed cell.**   We assume that stressed cells can communicate and trigger the conversion of a healthy to a stressed cell. We assume that there is a characteristic time scale $\beta$ associated with this process and the process is dependent on the presence of other stressed cells in the healthy cell's neighborhood.

We pick two different random numbers for this process — (a) A random number $r_1$ is first chosen for each healthy hepatocyte to see if its lesser than the conversion parameter $P_c = \Delta t/\beta$. If the condition is satisfied, the cell is added to a queue, (b) A second number $r_2$ is chosen and compared to the total number of stressed neighbors ($N_s$) in its neighborhood. A cell is

converted to a stressed type if $r_2 \leq N_s/T$, where $T$ is the total number of neighbors. Previously, majority rules, where the fate of each cell depends on the state of the majority of the neighbors have been also used to model opinion dynamics [48,49]. For 1st order neighbors, we consider 2 nearest cells in 1D and 6 nearest cells in 2D hexagonal lattice.

A 2D schematic of the rules is listed in Fig 1D–1F.

At the end of our simulations, we classify any case that recovers all it's healthy population ($H = N$, $S = 0$, $D = 0$) as recovery of the tissue and any simulation that results in all dead states as one of tissue death ($H = 0$, $S = 0$, $D = N$).

## Results

### (1) 1D pericentral necrotic damage and recovery

We first examined hematoxylin-eosin (H&E) stained liver tissue from mice sacrificed 24 hours after IP administration of APAP at 0, 10, 100, 250 and 500 mg/kg for evidence of hepatotoxicity. Necrotic zones are characterized by cytoplasmic vacuolization, loss of hepatic cytoplasm and nuclei, and congestion. In control sections and at low doses of APAP no discernable necrosis was present (Fig 2A–2C). At higher doses (Fig 2D and 2E) widespread centrilobular necrosis was evident. We also measured serum levels of the liver enzymes, alanine aminotransferase (ALT) and aspartate aminotransferase (AST), which are common clinical biomarkers of liver injury. Leakage of ALT/AST into the bloodstream occurs following hepatocyte injury and high levels can be indicative of significant necrosis. At 10 mg/kg and 100 mg/kg we saw modest increases in both ALT and AST (Fig 2F) even though no necrosis was evident by H&E (Fig 2B and 2C). At 250 and 500 mg/kg APAP we saw striking increases in both ALT and AST (Fig 2F), accompanying the widespread centrilobular necrosis seen by H&E (Fig 2D and 2E). This is in agreement with a detailed study by Roberts et al. (18) where they documented a dose-related increase in extent and area of centrilobular necrosis, accompanied by increases in centrilobular APAP adduct localization, and of GSH and ALT in serum. Further, by 48 hours mice recover from 250 mg/kg APAP, while 500 mg/kg is lethal. The questions that this suggests are: Given this damage pattern, what cell behaviors might underlie recovery in one case and death in the other; and what might be the outcomes of changes in extent of necrosis?

We then examined the time course of hepatocyte damage and recovery at 250 mg/kg APAP by H&E (Fig 2G–2K). Signs of stressed cells are seen by 2 hours after APAP, characterized by the appearance of hepatocytes with lightened cytoplasm around the CV (Fig 2H). Nuclei were still present in these stressed cells and appeared normal in size and morphology. Analysis of the initial damage pattern at 2 hours showed that the stressed area extended 32.6% of the distance along the CV to PT axis with a standard deviation of 16%. ALT/AST levels unchanged from control at this point indicating that hepatocytes were still intact (Fig 2L). Necrosis was progressive over time. By 4 hours necrosis was visible as lightened cytoplasm and condensation and loss of hepatic nuclei and was accompanied by congestion of the sinusoids with red blood cells (RBCs) around the periphery of the necrotic region. Necrosis was more prominent by 6 hours, with extensive loss of hepatic nuclei and continued RBC congestion. At both 4 and 6 hours after APAP treatment, necrosis was accompanied by an increase in ALT/AST, with the peak value occurring by 6 hours. The ALT/AST levels remained elevated and hepatocytes around the CV still appeared damaged at 24 hours. By 48 hours ALT/AST levels were near normal and the tissue had recovered with centrilobular hepatocytes appearing similar to the unaffected periportal cells. Taken together, the H&E data (Fig 2) gives us spatiality and a time scale for when cells become stressed (2 hours after APAP), have undergone necrosis (6 hours after APAP) and when the damaged centrilobular region is on the path to recovery (48 hours after APAP).

Resident and recruited Kupffer cells are activated by 2 hours after an APAP overdose [31,50]. Kupffer cell activation is accompanied by their release of pro-inflammatory cytokines that are largely cytoprotective but can also mediate cell damage [31]. We quantified the levels and localization of resident and recruited Kupffer cells after APAP treatment, using F4/80 labeling (Fig 3A–3E). We analyzed multiple lobular regions ($n$) around the CVs for a single animal; $n$ = 9 (0 hours), 14 (1 hours), 7 (2 hours), 7 (4 hours), 6 (6 hours), 5 (12 hours), 4 (48 hours) and 12 (72 hours). In untreated mice there were very few Kupffers and those that were present were rarely found in the centrilobular region (Fig 3A, 3D and 3E). Numbers and distribution of Kupffer cells remained unchanged from control through 4 hours after APAP treatment (Fig 3D and 3E). By 6 hours after APAP, Kupffer cell numbers had increased and they were localized primarily within the fourth cell layer out from the CV, at the periphery of the damaged centrilobular area, at the boundary between healthy and necrotic tissue (Fig 3B, 3D and 3E). This coincides with the worsened necrosis and peak ALT/AST levels also seen at 6 hours after APAP treatment (Fig 2). Proximity of the Kupffer cells to the edge between healthy and necrotic tissue suggests that any healthy cell at this location is at a greater risk of being affected by the damage. Kupffer cell numbers increase over the next 42 hours and their localization progresses pericentrally as the necrotic tissue is repaired (Fig 3C–3E), evidenced by the reduced ALT/AST levels at 48 hours after treatment. By 72 hours after APAP the Kupffer cells have returned to control levels and distribution (Fig 3E).

We examined changes in cell proliferation levels and localization using phosphohistone H3 labeling (Fig 3F–3J). We performed our analysis for a single animal at each of 6 time points with multiple ($n$) regions sampled; $n$ = 10 (0 hours), 9 (1 hours), 9 (2 hours), 8 (4 hours), 8 (6 hours) and 8 (48 hours). In control liver sections dividing cells are sparse and not concentrated in either periportal or pericentral regions (Fig 3F, 3I and 3J). At 1, 2, and 4 hours after APAP proliferation levels and localization are unchanged from control (Fig 3I and 3J). By 6 hours after APAP increased proliferation is evident and is concentrated at the periphery of the centrilobular region of necrotic cells (Fig 3G, 3I and 3J). The localization of dividing cells at 6 and 48 hours after APAP correlates with the leading edge of tissue recovery as it progresses toward the CV (Fig 3G–3J). This localization of hepatocyte proliferation during recovery of centrilobular necrosis being predominantly at the margin of the necrotic region supports our model assumption that loss of contact inhibition contributes to the increase in hepatocyte proliferation (Figs 2 and 3).

Based on our experimental results with APAP, we next show results for a CA model with initial conditions around 2 hours, characterized by the first appearance of stressed cells (Fig 2). On assuming a hepatocyte width of 30 μ$m$ and dividing the average CV to PT distance, we estimate 10 hepatocytes to be present in this region. We run our base model by setting $N$ = 10 with 40% of these hepatocytes to be stressed, i.e, $H_i$ = 6, $S_i$ = 4, $D_i$ = 0, since the H&E slides show approximately 30-40% of the CV to PT region as stressed. The number of healthy or stressed cells and the size of the tissue are all parameters in our model and we also make observations on varying them. We use a timestep of $\Delta t$ = 0.25 (hours) for the simulations and unless otherwise stated, all of our results are performed over 100 trial replicas. Our results explore the parameter space in terms of ratios of the timescales, by fixing any one timescale to a value and incrementing the other two timescales in steps of 0.50 (hours). This allows us to make general comments based on the output of the simulations in terms of biology of the system.

## (2) Small $\alpha$ combined with a large $\gamma$ leads to low recovery probabilities

In the 1D model, we identify two steady states for our simulations. $H = N$, $S = 0$, $D = 0$ is reached when the healthy states cells grow and the cells all completely fill up the simulated

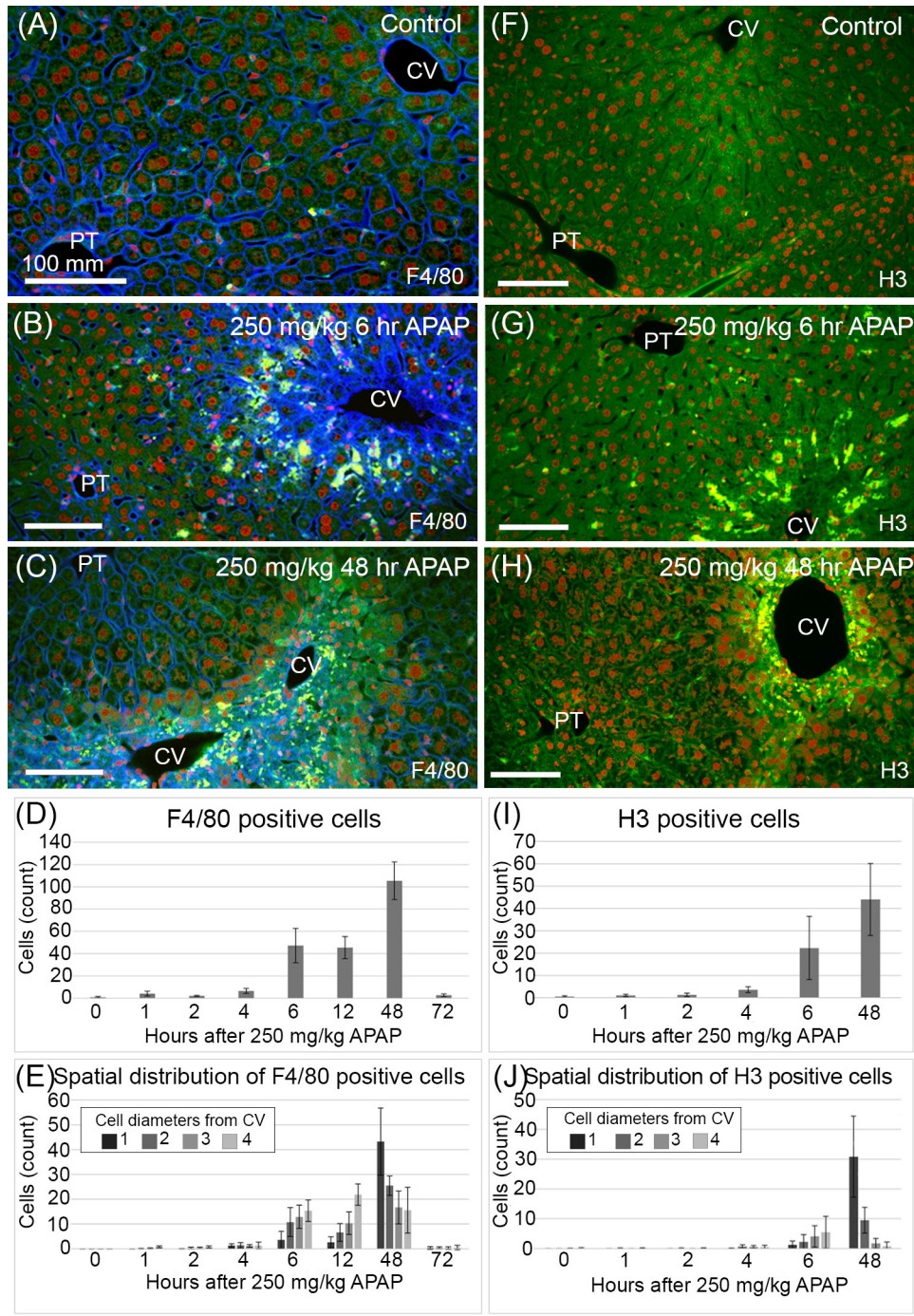

**Fig 3. Levels and localization of resident and infiltrating Kupffer cells and cell proliferation after APAP induced centrilobular necrosis.** (A-C) F4/80 labeling (yellow) identifies resident and infiltrating Kupffer cells. (F-H) H3 labeling (yellow) identifies proliferating cells. (A-C, F-H) Sections were counterstained to visualize cells using Lens Culinaris Agglutinin (LCA, green), nuclei using Sytox Green (red) and additionally in (A-C) plasma membranes of endothelial cells and hepatocytes using Tomato Lectin (TL, blue). The TL label also allows visualization of the prominent intracellular membrane bound regions in hydropic hepatocytes. (D,E) and (I,J) F4/80 and H3 positive cell counts are expressed as mean ± s.d. per 100,000 $\mu m^2$. (D) Macrophage counts greatly increase by 6 hours after APAP. Cell counts increase further by 48 hours after APAP. Kupffer cell counts return to normal by 72 hours after APAP. (E, J) Labeled cells (yellow) are counted within 1,2,3, and 4 cell diameters from central veins. (E) Kupffers are concentrated at the periphery of the necrotic region (4 cell diameters from the CV) at 6 and 12 hours after APAP. By 48 hours after APAP, the centrilobular damaged region has been almost filled with hepatocytes and macrophages are concentrated at

the CV (within 1 cell diameter from the CV). (J) Dividing cells are concentrated at the periphery of the necrotic region at 6 hours after APAP. By 48 hours after APAP dividing cells are concentrated near the CV.

tissue. The other is $H = 0$, $S = 0$, $D = N$ when all the healthy cells have been converted to a stressed state and then die. Here, averaging $H/N$ over the number of replicas gives a measurement of the probability of the tissue survival for a parameter combination.

Fig 4A shows the parameter space of $\gamma/\beta$ and $\alpha/\beta$ for a fixed value of $\beta = 5$ (hours). We find that the probability of the tissue surviving decreases with increasing ratios of $\gamma/\beta$. This can be understood as follows - if $\gamma$ is small compared to $\beta$, stressed cells die rapidly before influencing the adjacent healthy cells. As $\gamma$ increases, the long life of the stressed cells leads to greater damage to healthy cells. However this increase in $\gamma$ does not lead to a sharp transition to the steady state $H = 0$, $S = 0$, $D = N$, but the stochastic model instead shows regions of bistability in which the tissue could either survive or die. These bistable regions also show a gradient of output with the tissue recovery probability steadily decreasing as $\gamma$ increases (Fig 4A and 4B).

We define a criteria to mark the transition into the bistable region by tracking $\gamma/\beta$ and $\alpha/\beta$ values in which the probabilities of survival fall from 1 to 0.95 (points colored in red in Fig 4A). We find these points by finding the first occurrence of $\gamma/\beta$ for every $\alpha/\beta$ for which the survival probability is close to 0.95±0.01. The trend followed by the points indicate that this transition in 1D is almost independent of the proliferation timescale $\alpha$, at large values of $\alpha$, and can be approximately described by a critical ratio of $\gamma/\beta^* = 2.25\pm0.37$ (red line in Fig 4A). Sample outputs showing some of the trajectories along $\alpha/\beta = 1$ for different $\gamma/\beta$ ratios are shown in S1 Fig.

For smaller timescales of proliferation, the parameter space shows variation in the survival probabilities as a function of $\gamma/\beta$ and this behavior is more prominent beyond our critical ratio of $\gamma/\beta^*$. Fig 4B shows the survival probabilities as a function of $\gamma/\beta$ for $\alpha/\beta = 0.25, 3.25, 5.25$ shown in blue, black and magenta respectively. Interestingly, we find that that for the same value of $\gamma/\beta$, the probability of the tissue surviving is actually lowest for a small (fast) timescale

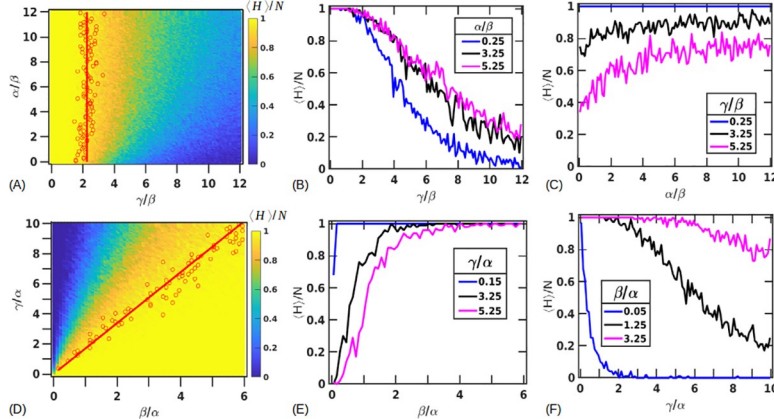

**Fig 4. Parameter space of tissue survival.** (A) Probability of tissue recovery as a function of $\gamma$, $\alpha$ with fixed $\beta = 5$ (hours). Metric plotted is the average fraction of healthy cells at steady state over 100 replicas, $\langle H \rangle/N$, using the colorbar as shown. Bistable states of survival are seen where the tissue sometimes survives ($H = N$, $S = 0$, $D = 0$) and sometimes dies ($H = 0$, $S = 0$, $D = N$). A critical ratio of $\gamma/\beta^* = 2.25\pm 0.37$ marks a transition (indicated by the red line) above which recovery probability falls down below 0.95. (B-C) Showing the variation in recovery probability by keeping $\alpha/\beta$ fixed at 0.25, 3.25, 5.25 (lines in blue, black, and magenta respectively) across values of $\gamma/\beta$ in (B) and by setting $\gamma/\beta = 0.25, 3.25, 5.25$ (lines in blue, black, magenta respectively) across values of $\alpha/\beta$ in (C). (D) Probability of tissue survival as a function of $\beta$, $\gamma$ with the value of fixed $\alpha = 5$ (hours). Red line indicates a ratio of $\gamma/\beta^* = 1.69\pm0.24$. Survival probabilities for fixed values of $\gamma/\alpha = 0.15, 3.25, 5.25$ (blue, black and magenta, respectively) across values of $\beta/\alpha$ (E) and fixed values of $\beta/\alpha = 0.05, 1.25, 3.25$ (blue, black and magenta, respectively) across values of $\gamma/\alpha$ (F).

of proliferation. We interpret this as follows – in this region ($\gamma/\beta > \gamma/\beta^*$) the conversion from a healthy to a stressed cell is more rapid than the clearance of the stressed cells. Cells will proliferate till they are limited by the number of spaces and the slow clearance of stressed cells. If proliferation is fast, healthy cells rapidly provide a domain for the stressed pattern to propagate into, as opposed to the inability of stressed cells to propagate into stressed or dead regions. This can also be seen in the S2 Fig that shows two different trajectory trials for a small proliferation timescale $\alpha/\beta$ = 0.05 (S2A and S2B Fig) and a larger proliferation timescale of $\alpha/\beta$ = 1.05 (S2C and S2D Fig). Both are for a large (slow) death timescale with $\gamma/\beta$ = 8.05. For the smaller proliferation timescale, we see the healthy population, shown in blue, peaking first, indicating rapid proliferation, that then begins to decay as the stressed cells take over and kill the healthy population. On the other hand, the few trajectories that survive for $\alpha/\beta$ = 1.05 don't usually show signs of increased proliferation till after the stressed population has decayed. Fig 4C also shows this behavior for three different values of $\gamma/\beta$ = 0.25, 3.25, 5.25 (blue, black, and magenta respectively) indicating that this behavior is only amplified for larger $\gamma$; the magenta curve shows a lower survival probability for small $\alpha$ before saturating to a probability of about 0.6.

We also plot the parameter space for $\beta$ vs $\gamma$ by fixing $\alpha$ = 5 (hours) (Fig 4D). As expected from Fig 4A, the transition to the bistable region is approximately described by a linear function with a constant ratio of $\gamma/\beta^*$. By averaging over $\gamma/\beta$ ratios where the survival probability of the tissue falls to 0.95, we obtain $\gamma/\beta^*$ = 1.69±0.24 (close to the ratio obtained in the previous case), indicated by the red line. For very low $\gamma$, we see that the probability of tissue survival is very high for all values of $\beta$ as the stressed cells die rapidly before being able to affect other cells. As $\gamma$ increases the bistability region expands leading to fewer survival chances for the tissue as a function of $\beta$. Fig 4E shows this for fixed value of $\gamma/\alpha$ = 0.15 (blue), 3.25 (black), 5.25 (magenta) across values of $\beta/\alpha$. Similarly, Fig 4F shows the survival probabilities for fixed values of $\beta/\alpha$ = 0.05 (blue), 1.25 (red), 3.25 (orange) across values of $\gamma/\alpha$. Parameter spaces with fixed $\gamma$ and varying $\beta$ and $\alpha$ is shown in S3 Fig.

## (3) Proliferation profiles show peaks in the number of new hepatocytes

Differences in the survival probabilities at low $\alpha$ can also be attributed to the regenerative capacity of the different cell states for certain parameter combinations. To better understand this, we quantify parameter regions where the maximum number of cell divisions occur, creating new hepatocytes, as the simulation regains the total liver mass. Fig 5A shows the average number of new hepatocytes created during the simulation for fixed $\beta$ = 5 (hours), $N$ = 10, $H_i$ = 6, $S_i$ = 4. For very small $\gamma$, since stressed cells die quickly, the new healthy cells created are the only ones needed to fill the number of spaces left by the stressed cells. For large $\gamma$, due to a longer lifetime of stressed cells, all the healthy cells are eventually converted to stressed cells, which limits the number of cell divisions. In between these two cases of small and large $\gamma$, we see a peak in the number of new hepatocytes created. This peak is more prominent at smaller proliferation timescales. Additionally, since smaller proliferation timescales also result in a greater probability of tissue death, we see a decrease in the number of new hepatocytes created for small $\alpha$ and large $\gamma$ (Fig 5A and 5C). Thus, smaller proliferation timescales produce a larger peak indicating a large regenerative capacity for moderate values of $\gamma$, before declining more for large values of $\gamma$, as compared to large values of $\alpha$. Additionally, the line in Fig 5A indicates $\gamma/\beta^*$ = 2.25±0.37, denoting the ratio of $\gamma/\beta$ above which the survival probability falls below 0.95. We note that the peak of maximum divisions is slightly to the right of this line, suggesting that in some cases even having the ability to regenerate many times isn't still sufficient to stop the tissue from dying. Simulations with larger $N$ show the same behavior with now the $\gamma/\beta^*$ and the peak of maximum proliferation shifted to the right (S4 Fig).

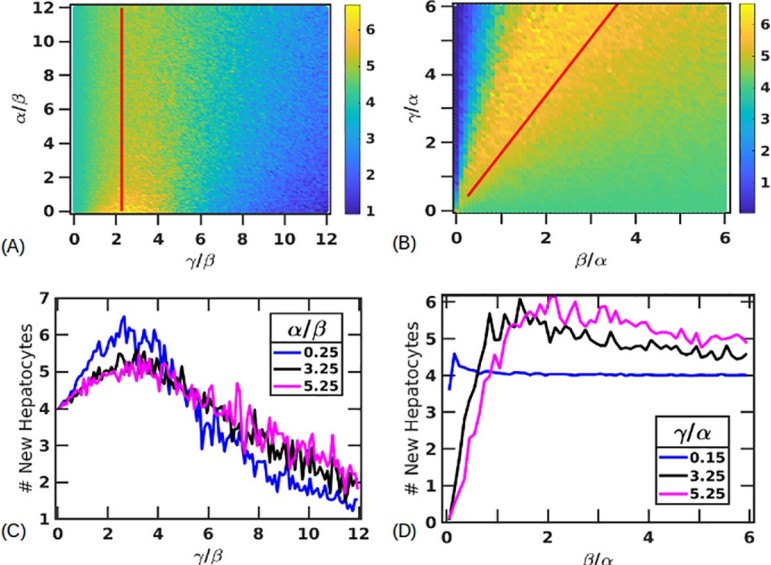

**Fig 5. Number of new hepatocytes created.** Average number of new hepatocytes (shown by the colorbars) created for fixed $\beta = 5$ (hours) (A, C) and fixed $\alpha = 5$ (B, D). Lines indicate $\gamma/\beta^* = 2.25\pm0.37$ in (A), and $\gamma/\beta^* = 1.69\pm0.24$ in (B). (C) shows this as a function of $\gamma/\beta$ for fixed $\alpha/\beta = 0.25$ (blue), 3.25 (black), 5.25 (magenta) and (D) as a function of $\beta/\alpha$ with fixed $\gamma/\alpha = 0.15$ (blue), 3.25 (black), 5.25 (magenta).

Fixing $\alpha = 5$ (hours) shows a region characterized by $\gamma/\beta$ ratios for which maximum proliferation can be now seen (Fig 5B). Horizontal lines across this plane are shown for fixed $\gamma/\alpha = 0.15$ (blue), 3.25 (black) and 5.25 (magenta) (Fig 5D).

## (4) 1D critical threshold of damage pattern propagation

Our in vivo mouse dose response studies indicated that APAP at high doses (500 mg/kg) is often fatal by 48 hours, presumably due to massive liver necrosis, while animals treated with a lower dose of 250 mg/kg recover. Here we ask the question - Is there an extent of tissue damage beyond which damage is fatal and unrecoverable? To do this, we look at the minimum number of healthy cells $min(H)$ that simulations can reach from an initial number of healthy cells ($H_i$) and still eventually survive. We quantify this loss of healthy cells as a function of the parameters with

$$Loss = (\langle min(H) \rangle - H_i)/H_i * 100\%$$

We make our observations with a fixed $\alpha$ and quantify our results in terms of $\gamma/\beta$ ratios.

Fig 6A shows how the tolerable extent of damage is dependent on $\gamma/\beta$ (colorbar shows the loss). For tissues that have a survival probability of 0.95 and above, the maximum damage is seen to be around 20% as $\gamma/\beta$ approaches 2.0. As $\gamma/\beta$ is increased even further, much larger extents of damage can cause lower survival probabilities. However, parameters set with a very high ratio of $\gamma/\beta$ showed reduced signs of damage for simulations that progressed to survival of the tissue. This is because under these parameter combinations, the simulations that do survive must only sustain minimal losses from the initial number of healthy cells.

Alternatively, we can plot the maximum number of healthy cells the simulations can gain and still have the simulation progress to all dead cells. We define a gain fraction as $(\langle max(H) \rangle - H_i)/H_i * 100\%$ and the results are shown in S5 Fig. We find that very few healthy cells are gained in simulations that ultimately progress to complete cell death. The average gain for

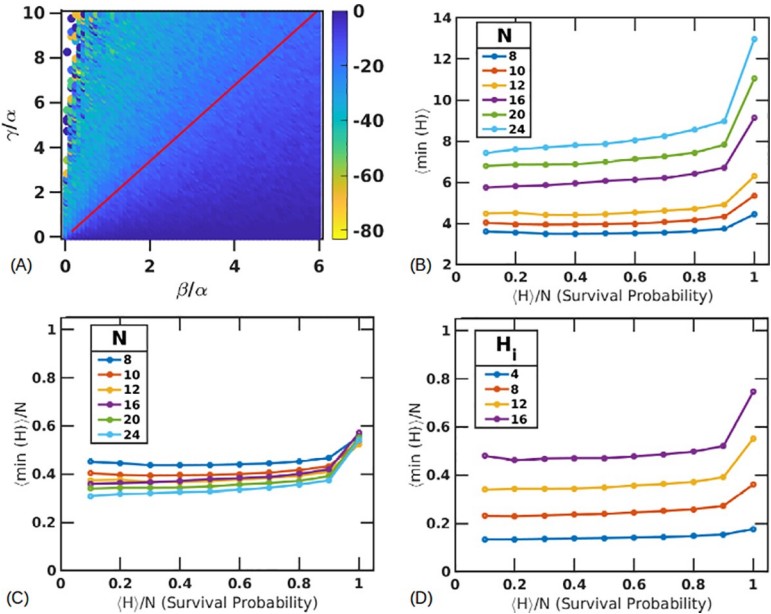

**Fig 6. Critical thresholds of healthy cell populations.** Loss of healthy tissue resulting in a second wave of damage, as a function of the parameters $\gamma$ and $\beta$ for fixed $\alpha$ = 5 (hours). Statistics are measured only for simulations where the cell population recovers and the red line shows $\gamma/\beta$ = 1.69 (A). Average size of the minimum healthy population $\langle min(H)\rangle$ plotted against final length $\frac{\langle H\rangle}{N}$ (B). $\langle min(H)\rangle$ is now rescaled against the tissue length ($N$) and plotted against $\langle H\rangle/N$ (C). $\langle min(H)\rangle/N$ plotted with different lengths of initial damage (D).

across the entire studied parameter range is below 10%. The gain for high $\gamma/\beta$ is low, as damage is easily propagated due to the longer presence of stressed cells. The gain in cells increases as $\gamma/\beta$ decreases, before falling again as $\gamma/\beta^*$ is reached where stressed cells are now more easily eliminated. There are very few simulations that progress to complete cell death in this region. A few outliers are seen, characterizing lone trials where this gain is as large as 50%.

Next, while this loss or gain percentages are parameter dependent, we also checked if the tissue shows a critical size of healthy populations. We do this by plotting $\langle min(H)\rangle$ as a function of the binned final average tissue length, $\langle H\rangle/N$. Since this final length also represents the probability of the tissue surviving, this will give us a correlation on how likely it is for the simulation to be successful after reaching a certain minimum healthy cell population. We find that with $N$ = 10, and $H_i$ = 6, $S_i$ = 4 (initial damage of 40%), the average minimum length in simulations that survive saturate to around 4 healthy cells, indicating a lower threshold for parameter combination with large $\gamma/\beta$ ratios (Fig 6B). Additionally, we find that this lower threshold is dependent on the tissue size ($N$). Varying the size of the tissue with different $N$ and the same damage fraction (around 40%) shows different critical lengths for the minimum number of healthy hepatocytes. However, scaling these trends by $N$ shows that the critical length is always the same for the same initial amount of damage fraction (Fig 6C). In this case, this fraction is 0.37±0.04, indicating that if the healthy population drops below about 37% of the total population, the damage is likely to progress to complete cell death. However, this critical length is dependent on the initial amount of damage as shown in Fig 6D. Here the different lines are plotted with the same $N$ (20) for $H_i$ = 4, $S_i$ = 16 (blue), $H_i$ = 8, $S_i$ = 12 (red), $H_i$ = 12, $S_i$ = 8 (orange) and $H_i$ = 16, $S_i$ = 4 (purple). Fig 6D shows that the presence of fewer initial stressed cells can accommodate greater losses to the healthy populations indicated by a $\langle min(H)/N$ of 0.48 ($H_i$ = 16, $S_i$ = 4), 0.34 ($H_i$ = 12, $S_i$ = 8), 0.23 ($H_i$ = 8, $S_i$ = 12) and 0.13 ($H_i$ = 4, $S_i$ = 16).

## (5) Critical ratio $\gamma/\beta^*$ is proportional to the size of the tissue

The previous simulations indicated that a minimum threshold is dependent on the size of tissue. We additionally quantify how the phase spaces behave as we vary the number of initial stressed cells and the total size of the tissue. On changing $H_i = 2$, $S_i = 8$ (initial damage at 80%) we find that the region where the tissue survives is reduced drastically (Fig 7A). Approximating the boundary where the tissue survival (and the average length) falls to 0.95±0.01, we measure a slope ($\gamma/\beta^*$) of 0.38±0.11. This slope value increases to 1.07±0.18 for $H_i = 4$, $S_i = 6$ (initial damage at 60%) (Fig 7B) and 2.53±0.35 for $H_i = 8$, $S_i = 2$ (initial damage at 20%) (Fig 7C). These results again show how the probability of recovery is dependent on the initial length of damage in the 1D model.

Similarly, we can vary the total length of the tissue ($N$), simulating the highly variable CV-PT distance observed in liver lobules, and evaluate how the initial damage pattern now affects the survival probability of the tissue. We found that for a given fraction of initial damage, a larger tissue size expands the parameter region where the tissue survives (Fig 7D–7F). Here we have fixed the initial pattern of damage near 40% for different $N = 8$ ($H_i = 5$, $S_i = 3$) (Fig 7D), $N = 16$ ($H_i = 10$, $S_i = 6$) (Fig 7E) and $N = 24$ ($H_i = 14$, $S_i = 10$) (Fig 7F). Approximate slopes determined are $\gamma/\beta^* = 1.46\pm0.26, 2.74\pm0.28, 3.87\pm0.74$ respectively.

We can see the combined trends in Fig 7G, with the critical ratio $\gamma/\beta^*$ plotted on the y axis. We find that rescaling all the curves by the length of the tissue essentially collapses them (Fig 7H), showing that $\gamma/\beta^*$ is dependent on the tissue size. For 40% initial damage, the approximate value of $\gamma/\beta^* = (0.17\pm0.02)N$.

## (6) 2D results show coexistence of states

We next implement the CA rules on a 2D hexagonal lattice modeling a section through a liver lobule with CV-PT length ($N_L$) of 10 ($N = 330$) (S6 Fig). Our initial conditions have four layers

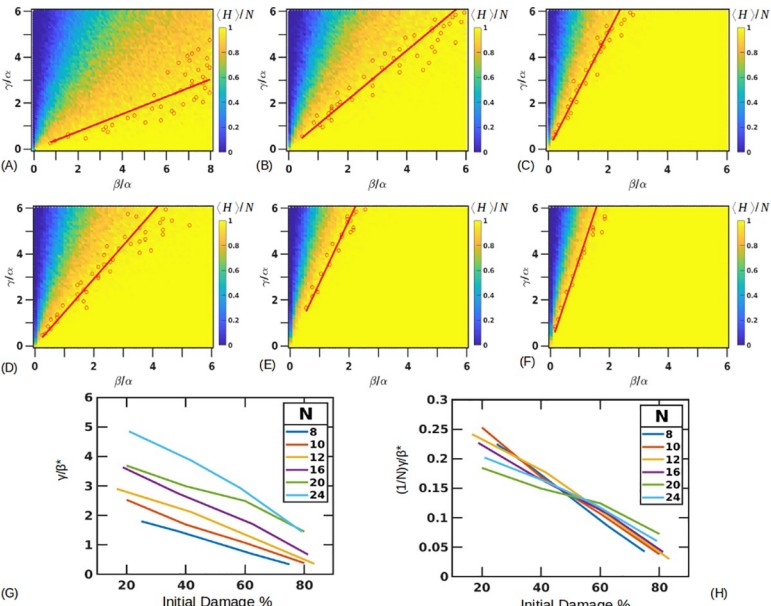

**Fig 7. Variation of the initial damage pattern and tissue size.** (A-C) Fixed tissue length ($N = 10$) with $H_i = 2$ (A), $H_i = 4$ (B) and $H_i = 8$ (C) with $\alpha = 5$ (hours). Lines are drawn through points where survival probability drops to 0.95 ±0.01. Values of $\gamma/\beta^* = 0.38\pm0.11, 1.07\pm0.18, 2.53\pm0.35$ (A-C). (D-F) Fixed damage fraction ($\cong$40%) with different tissue lengths of $N = 8, 16, 24$. Approximate slopes are $\frac{\gamma}{\beta^*} = 1.46 \pm 0.26, 2.74 \pm 0.28, 3.87 \pm 0.74$. (G) Combined curves showing the slope boundaries for different fractions of damage with different tissue lengths. (H) Rescaling the slopes by the tissue lengths collapses the curves.

of stressed cells centered at the CV to represent 40% damage along the CV-PT axis. We run our parameter scans for a fixed $\beta = 5$ (hours). Additional examples of exploring the phase space for different values of $\alpha$ is shown in S7 and S8 Figs. We find that now we have a richer phase space comprising solutions in which the tissue fully recovers ($H = N$, $S = 0$, $D = 0$), where the tissue completely dies ($H = 0$, $S = 0$, $D = N$) and a "coexistence state" where neither the healthy nor the dead population take over but instead the three cell types coexist indefinitely ($H^*$, $S^*$, $D^*$). This third state appears to be caused partly by cells having a larger number of contact neighbors (six in 2D versus two in 1D) which provide additional information to the cell, and partly from parameters being in regions of maximum regenerative capacity (Fig 5). Fig 8A shows the phase space of all the outcomes from the trials. In Fig 8A, any parameter combination over all the trials that leads to the complete recovery of healthy cells is colored yellow (marked as S), simulations where all the cells die are colored green (marked as D) and simulations where coexisting populations are obtained are shown in red (marked as C). We note that the transitions between these different phase regions are not sharp, but rather gradual as indicated in Fig 8A, apparently because of the stochasticity of the model. Unlike the 1D case, the 2D case does not predict a single $\gamma/\beta^*$ ratio for the transition from recovery to the different phase outcomes. However, we note that signs of tissue death in Fig 8A begin to appear at an approximately constant ratio of $\gamma/\beta = 5.03 \pm 0.30$.

Fig 8B shows the average number of healthy cells at steady state for each parameter combination. For the coexisting states, this value is taken to be the final value after running the simulation for a very long time, on the time scale of $\approx 500$ days. Some sample trajectories for the parameter sets indicated by the dashed line in Fig 8B are shown in Fig 8C–8G with additional visualization output plotted in S6 Fig. For a small $\gamma/\beta = 0.05$ (Fig 8C and S6A Fig), stressed

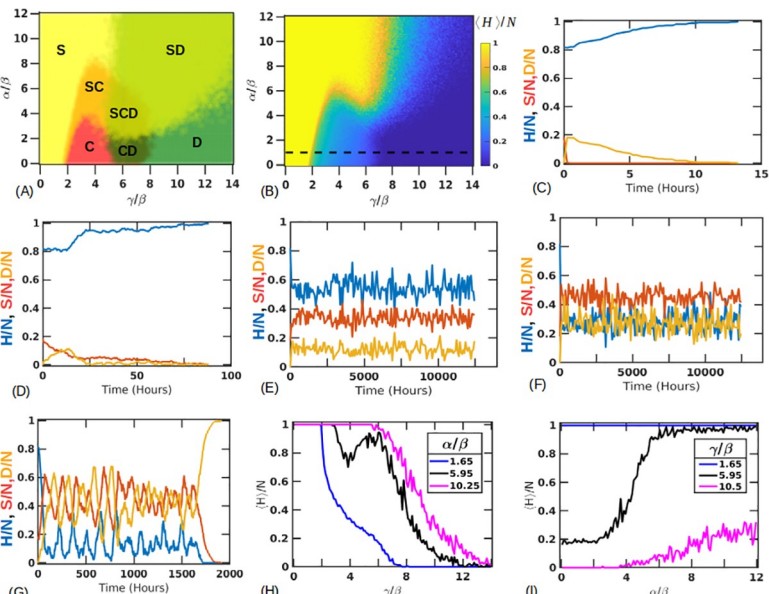

**Fig 8. 2D Parameter space of tissue repair.** (A) Simulations on a hexagonal lobule with fixed $\beta = 5$ (hours) show presence of coexisting states shown in red (marked with C). Other regions show parameters for which the tissue always survives (shown in yellow, marked with S) and regions where the tissue always dies (shown in green, marked with D). The other colors are combinations of these states where different outcomes are possible for the same parameter set. (B) Average size of the number of healthy states. Sample trajectories along the dashed line shown for $\gamma/\beta = 0.05$ (C), 1.45 (D), 2.45 (E), 4.45 (F), 9.05 (G). Coexisting states also show a gradient in their steady state population as shown for (E, F). (H) Average healthy population is plotted along fixed ratios of $\alpha/\beta = 1.65$ (blue), 5.95 (black), 10.25 (magenta) across $\gamma/\beta$. (F) Average healthy population is plotted along fixed ratios of $\gamma/\beta = 1.65$ (blue), 5.95 (black) and 10.5 (magenta) across $\alpha/\beta$.

cells die rapidly and healthy cells quickly take their place. As $\gamma/\beta$ is increased to 1.45 (Fig 8D and S6B Fig), stressed cells live longer affecting the tissue to a greater degree, but the tissue can still recover. On increasing $\gamma/\beta$ = 2.45, we reach a domain where stressed cells and healthy cells coexist without any of the populations ever winning out (Fig 8E and S6C Fig). The average number of healthy or stressed cells depends on the value of $\gamma/\beta$ ratio and as this value is increased to 4.45 (Fig 8F and S6D Fig), the coexisting populations have a higher stressed cell population compared to Fig 6E. The populations in both these cases seem to fluctuate around a steady state where the healthy, stressed, dead populations are $(H^*, S^*, D^*)$ = (0.54±0.06, 0.34 ±0.04, 0.12±0.04) (Fig 8E) and $(H^*, S^*, D^*)$ = (0.28±0.07, 0.46±0.05, 0.27±0.07) in Fig 8F. Additionally, these coexisting state populations are surprisingly stable to step perturbations in the distribution of cell types, as shown in S9–S11 Figs. However, adding a baseline noise ($\epsilon$) to our conversion rule, assigning a probability to cells becoming stressed even in the absence of other stressed cells, reduced the healthy population fractions for the same parameter sets of $\gamma/\beta$ = 2.45, 4.45. Here setting $\epsilon$ = 0.01, 0.05, 0.1 for $\gamma/\beta$ = 2.45 changed the average healthy population to $(H^*, S^*, D^*)$ = (0.51±0.06, 0.36±0.04, 0.14±0.04), $(H^*, S^*, D^*)$ = (0.42±0.05, 0.39±0.04, 0.19 ±0.05), $(H^*, S^*, D^*)$ = (0.36±0.06, 0.40±0.04, 0.24±0.06) respectively. A similar behavior was seen for $\gamma/\beta$ set to 4.45 which showed $(H^*, S^*, D^*)$ = (0.27±0.07, 0.45±0.06, 0.28±0.08), $(H^*, S^*, D^*)$ = (0.23±0.07, 0.44±0.07, 0.33±0.09), $(H^*, S^*, D^*)$ = (0.19±0.06, 0.42±0.07, 0.39±0.09) for $\epsilon$ = 0.01, 0.05, 0.1 respectively (S12 Fig).

As $\gamma/\beta$ is increased even further to 9.05 (Fig 8G and S6E Fig), the stressed cells persist longer and are able to affect other cells. This eventually pushes the tissue towards death.

Fig 8H focuses on the healthy steady state populations across different values of $\gamma/\beta$ for fixed $\alpha/\beta$ = 1.65(blue), 5.95 (black) and 10.25 (magenta). The first two lines cross the coexistence regions and a dip in the population numbers can be seen in the black line. This is interesting as it shows that coexisting states with a lower healthy population exist next to parameter regions showing a high degree of survival, before transitioning into bistable regions again. Additionally, as in the 1D case, fast proliferation appears to lead to a decreased healthy state population.

Similarly, Fig 8I focuses on the healthy state populations across different values of $\alpha/\beta$ for fixed $\gamma/\beta$ = 1.65 (blue), 5.95 (black) and 10.5 (magenta). As $\gamma$ increases (slower death timescale), the simulations show a lower healthy population.

We also find that any initial asymmetries in the distribution of stressed cells doesn't change the nature of the phase space outcome (S13 Fig). However, simulations with a larger number of cells ($N_L$ = 50, $N$ = 7650) showed expansion in the coexisting state space regime (S14A Fig). Varying the initial conditions in this case showed shifts in the SD boundary as in the 1D case with little changes in the localization of the coexisting states (S14B and S14C Fig).

## (7) 2D Space with second order neighbor effects

We explored the parameter and phase space of the 2D model in the presence of second order neighbor effects. In this case, a given healthy cell responds based on the state of both its six first order neighbors and on its 12 second order neighbors. This simulates the effect of a short range diffusible signal that increases the range of cell-cell communication from contact neighbors to neighbors separated by one intervening cell. We find that while the nature of the 2D phase space doesn't change greatly (Fig 9A), a shift in the boundary between the survival (S) and the bistable (SD) regions can be seen (compare Figs 8A and 9A). Additionally, at large $\gamma/\beta$, for the same parameter values as in Fig 8A, the tissue shows a small increase in the probability of tissue death. Fig 9B plots the healthy cell population and Fig 9C shows the healthy population fraction across $\gamma/\beta$ for fixed values of $\alpha/\beta$ = 1.65 (blue), 4.95 (black), 10.25 (magenta).

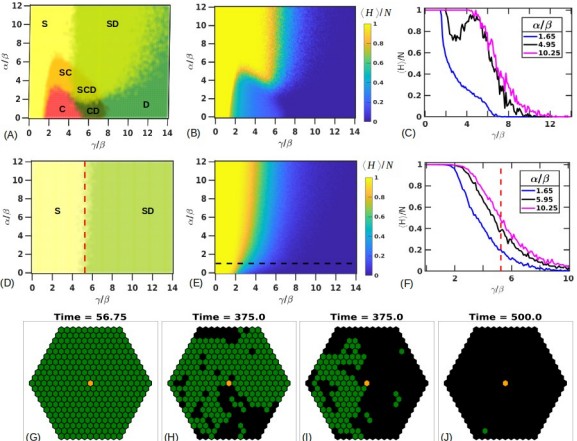

**Fig 9. Second order neighbor effects and fixing number of divisions.** (A) Simulations on a hexagonal lobule with fixed $\beta$ = 5 (hours), including second order neighbor effects. Coexisting states shown in red (marked with C). Other regions show parameters for which the tissue always survives (shown in yellow, marked with S) and regions where the tissue always dies (shown in green, marked with D). The other colors are combinations of these states where different outcomes are possible for the same parameter set. (B) Average size of the number of healthy states. (C) Healthy population fraction across $\gamma/\beta$ for fixed values of $\alpha/\beta$ = 1.65 (blue), 4.95 (black), 10.25 (magenta). (D) Simulations on a hexagonal lobule with fixed $\beta$ = 5 (hours) and fixing the number of maximum divisions per hepatocyte removes the coexistence states. Criteria of tissue survival is the presence of healthy cells at the end. Red dashed line showing the transition from the survival states at $\gamma/\beta^*$ = 5.25±0.29 (E) Average size of the number of healthy states. (F) Healthy population fraction across $\gamma/\beta$ for fixed values of $\alpha/\beta$ = 1.65 (blue), 5.95 (black), 10.25 (magenta) with the red dashed line showing $\gamma/\beta^*$ = 5.25±0.29. Visualization output showing a representative state at the final simulation time point for $\alpha/\beta$ = 1 and $\gamma/\beta$ = 1.45 (G), 2.45 (H), 4.45 (I) and 9.05 (J) at the final time.

## (8) 2D Coexisting states depend on the number of divisions

In the previous 1D and 2D simulations, each hepatocyte can divide as many times as needed. Actual cells may have a limited division potential. Hepatocytes are known to divide throughout the life of an individual, although some experiments suggest that the number of divisions per hepatocyte may be limited [51,52]. We find that fixing the number of maximum allowed divisions per hepatocyte to 1 removes the coexistence states (Fig 9D). However, including this limitation also limits the hepatocytes' ability to completely repopulate the liver lobule space. Therefore, we relax our criteria to denote our survival 'S' states as any simulation outcome that has healthy cells at the end without any stressed cells. Fig 9D indicates that we begin seeing signs of complete tissue death near $\gamma/\beta$ = 5.25±0.29, this is near the same ratio of $\gamma/\beta$ = 5.04 ±0.30 where we saw the first appearance of complete tissue death in simulations with no limit on the maximum number of divisions. Fig 9E shows the average healthy population as a function of $\gamma/\beta$ and $\alpha/\beta$. Fig 9F shows the healthy population levels across $\gamma/\beta$ for fixed values of $\alpha/\beta$ = 1.65 (blue), 5.95 (black), 10.25 (magenta) with the red dashed line showing $\gamma/\beta^*$ = 5.25 ±0.29, the value at which the system transitions from recovery to death. Fig 9G–9J show the visualization output for $\alpha/\beta$ = 1 and $\gamma/\beta$ = 1.45 (G), 2.45 (H), 4.45 (I) and 9.05 (J) for simulations with the cell limited to a single division. The outputs (G-J) shown are at stationary states since there are no remaining stressed cells and/or the remaining live cells have reached their proliferation limits. Comparing this to the output for Fig 8, we note that the parameter sets used in (H-I) previously showed signs of coexistence. S15 Fig shows that increasing the maximum number of allowed divisions to just two brings back the coexisting states and gives a model that begins to mimic the one shown in Fig 8, in which there was no limit on the number of times a cell can divide.

## Discussion

In this paper we have explored a CA based model of hepatocyte behavior in the liver following a drug induced centrilobular injury. Based on observations from our experimental data in mice, we have represented hepatocytes in three different states — healthy, stressed and necrotic/dead. The simulated cells change states based on their current and neighboring cells' states. Healthy cells can divide with a proliferation time scale of $\alpha$ to regain liver mass by replacing adjacent empty spaces left by dead cells. Healthy cells can also be converted into stressed cells due to interaction with neighboring stressed cells with a timescale of $\beta$. The stressed cells disappear (die) after becoming necrotic with a timescale of $\gamma$, leaving behind empty spaces. We model the process of response to an initial large scale tissue damage as a competition between these different processes.

To facilitate exploring the parameter space, we vary two of the parameters while keeping the third fixed and expressing the varied parameters as a ratio with the fixed parameter. By varying the ratios of any two timescales, we discover parameter combinations that lead to either tissue recovery or death. We find that the transition from recovery to death is not sharp but characterized by parameter sets where either outcome is possible thus showing probability gradients of tissue recovery. For larger values of the proliferation timescale $\alpha$, the probability of the tissue surviving can be described by the $\gamma/\beta$ ratios alone (Fig 4A). However, at smaller values of $\alpha$, variations in $\alpha$ become important with the probability of tissue death being the highest for smaller $\alpha/\beta$ ratios and large $\gamma/\beta$ ratios (Fig 4A and S2 Fig). We can characterize the parameters that show tissue death by approximating critical ratios of $\gamma/\beta^*$ beyond which the tissue has less than a 0.95 probability of survival. This $\gamma/\beta^*$ is dependent on the size of the tissue ($N$) and initial amount of damage (number of stressed cells). For an initial 40% damage, $\gamma/\beta^* \approx (0.17 \pm 0.02)N$ (Fig 7). Our CA model also predicts a minimum healthy population size of $0.37N$ for the initial 40% damage, below which the liver damage is likely irreversible (Fig 6).

Applying the same set of rules to a 2D model produces a richer parameter and phase space where additional coexisting states can be identified (Fig 8A). In the coexisting state, the cells persist in each of the three possible states without the system ever going to complete recovery or complete death. The coexisting states are stable to external perturbations that give rapid changes in the size of one of the three populations (S9–S11 Figs). These coexisting states are localized around regions of maximum proliferation (Fig 5) and limiting the number of divisions per hepatocyte to 1 removes these coexisting states (Fig 9). However, doing so also limits the hepatocyte's ability to completely fill the liver tissue. This suggests that other requirements need to be put in place, like an accompanying increase in the healthy hepatocyte sizes during recovery as seen in experiments [52].

This simple CA model makes interesting predictions on the parameter and phase spaces of hepatocyte behavior in a spatially defined context. Our model can be compared to biological parameter regimes by observing that the first signs of stressed cells appear around 2 hours and prominent necrosis appears over the next 4 hours. Previously reported literature values have also suggested that the hepatocyte doubling time is approximately 24 hours [53]. Based on these observations we can approximate our parameters as $a \approx 20$ hours, with $\beta \approx 1-2$ hours and $\gamma \approx 2-4$ hours. These values give us a range of ratios of $\beta/\alpha = 0.05-0.1$ and $\gamma/\alpha = 0.1-0.2$. For $H_i = 6$, $S_i = 4$ (initial damage of 40%), the recovery probabilities as seen in the 1D simulation are 0.88 (Loss = -28.5%) and 0.97 (Loss = -18.8%) for $\beta/\alpha$, $\gamma/\alpha = 0.05$, 0.1 and $\beta/\alpha$, $\gamma/\alpha = 0.1$, 0.2 respectively. Similarly, for $H_i = 7$, $S_i = 3$ (initial damage of 30%) the survival probabilities are 0.90 (Loss = -21.4%) and 0.99 (Loss = -16.9%). 2D simulations for the same parameters show recovery for all the simulations for the parameter sets with losses of -25.7% and -13.7% ($H_i = 270$, $S_i = 60$, 4 stressed layers along the CV-PT axis) and -14.7% and -0.03% ($H_i = 294$, $S_i = 36$,

3 stressed layers damage along the CV-PT axis). The high probability values at these ratios indicate that the tissue will recover under these conditions. This is also seen in our histology slides. Additionally, S16 Fig showing the normalized damaged fraction at the different time points from H&E slides indicate that the fraction remains relatively unchanged, with no large losses to the healthy cells, as is also seen in these simulations. However, this analogy is limited somewhat by the limited number of time points (four) in the in vivo studies. Future mouse models with a finer time resolution of spatial necrosis patterns could help test these predictions based on the three suggested parameters. Additionally, we can also compare the time estimate of tissue recovery after the APAP treatment by adding an initial 2 hours for time taken to reach the steady state in our simulations. For our 1D simulations, for the chosen sets as above, this estimate is around 25-30 hours while for the 2D simulations it is 22-25 hours. This average value of around a day is lower than recovery period of >48 hours as is seen in the H&E slides. However, since this time is heavily dependent on the value of $\alpha$, setting the time for how fast the healthy cells proliferate, a better estimate of this value will be able to approximate the model behaviour better. Finally, while we don't expect an additional recovery timescale to change our results qualitatively, we expect that the boundaries of the different domains would shift, which could also lead to finer predictions for the experiments.

While these results have been based on a pipe or a planar layout, these simulations can also be extended to a 3D framework where cells are in contact with additional neighbors in planes above and below making the total number of neighboring sites 8. Compared to the 2D case, these simulations show that the coexistence regions expand indicating that the neighbor interaction range could be an important parameter (S17 Fig).

In conclusion, we have developed stochastic CA models of progression of injury in liver lobules following a pericentral toxic challenge. The modeling delineates the parameter space and resultant tissue response phase space including complete tissue recovery, complete tissue death and situations where the tissue appears to settle into a partially damaged state that persists indefinitely, as long as there is sufficient proliferative capacity from existing hepatocytes. This partially damaged persistent state, though probably not relevant to APAP toxicity, may be relevant to chronic liver disease such as hepatitis and alcoholic cirrhosis. Our results also show that rapid proliferation does not always ensure tissue recovery, indicating possible situations where proliferation might not be able to limit liver damage. Biological perturbations to our timescales, aided by a hepatocyte's interaction range, will be able to test out these predictions, including the presence of critical ratios of tissue recovery and the dependence on a minimum healthy hepatocyte population beyond which damage could be irreversible. Regions in the parameter space where it is not possible to predict the outcome of the simulation would be of particular interest as they represent situations where the outcome might be critically dependent on small changes in other factors.

## Methods

### Experimental animal model

All animal experiments were approved and conducted according to Institutional Animal Care and Use Committee guidelines of Indiana University, and adhered to the NRC guide for care and use of animals. Twelve hours before APAP administration, 7-10 week old male C57BL/6 mice were moved to clean cages without food. The day of the study mice received an IP injection of APAP (dose of 250 mg/kg) or saline vehicle (at a volume of 0.2 mL/10 g body weight), and were then returned to cages with food. Mice were then analyzed over time.

**Liver enzymes.** Liver enzymes, alanine aminotransferase (ALT) and aspartate aminotransferase (AST), were measured immediately following serum separation from whole blood

using Infinity™ ALT and AST kits (Thermo scientific, Middletown, VA) with a UV microplate reader (Tecan Infinite M200Pro, San Jose, CA).

**Histopathology.** Following euthanasia, mouse livers were perfused with warm saline and removed and fixed in 10% neutral-buffered formalin. Fixed liver tissues were paraffin embedded, sectioned (4–5 $\mu$m), and stained by hematoxylin and eosin (H&E) for the evaluation of pathological changes.

## Fluorescent labeling

Following euthanasia, mouse livers were perfused with warm saline, then with warm 4% paraformaldehyde in PBS. Livers were then removed and incubated overnight in 4% paraformaldehyde in PBS. Fixed livers were vibratome sectioned (200 mm). Liver sections were blocked and permeabilized overnight using 0.5% Tween in Carbo-Free Blocking Solution (Vector Laboratories). Sections were then either incubated in anti-F4/80 antibody (Serotech) to label resident (Kupffer) and infiltrating macrophages [54] or anti-histone-H3 antibody (Invitrogen) to detect proliferation [55], followed by fluorescent secondary antibodies for visualization (Invitrogen). Morphological staining was applied to all sections using rhodamine-conjugated lens culinaris agglutinin (LCA, Vector Laboratories) to highlight cells and tissues and Sytox Green (Invitrogen) to label nuclei. In some sections cell membranes were also labeled using 633-conjugated Tomato Lectin (TL, Vector laboratories). Labeled sections were either mounted in PBS or optically cleared and mounted [56].

## Microscopy and image analysis

We acquired confocal images of the fluorescently labeled tissues using Leica SP2 confocal/MP and Leica SP8 confocal microscope systems. We conducted quantitative image analysis using FIJI [57,58]. Using the Cell Counter plugin, we counted F4/80 and phosphohistone H3 positive cells and classified them by distance in number of cell diameters from a central vein. To quantify the proportion of damaged cells in the H&E images, we used Color Deconvolution [59] to separate color channels followed by segmentation using Weka [60] using the intensity of the cytoplasmic staining to identify the transition from healthy to damage cell regions. Along a line from each CV to the nearest PT, we measured the proportion of damaged cells.

## Simulations

CA model was implemented using Python.

## Supporting information

**S1 Fig. 1D Visualization output at fixed $\beta$.** 1D simulation outputs for fixed values of $\beta$ = 5 (hours). All of the figures are for fixed $\alpha/\beta$ = 1and $\gamma/\beta$ = 0.05 (A), 1.45 (B), 2.45 (C) and 9.05 (D) with trajectories shown in (E). (A),(B) lie to the left of line demarcating $\gamma/\beta^*$ = 2.25±0.37 and have more than 0.95 probability of regaining all the healthy cells.
(TIF)

**S2 Fig. Trajectories with different proliferation timescales.** Trajectories for a small (fast) proliferation timescale $\alpha/\beta$ = 0.05 for two different trials (A,B) and a larger (slower) proliferation timescale of $\alpha/\beta$ = 1.05 (C,D). Both are for a large (slow) death timescale with $\gamma/\beta$ = 8.05 and show how a fast proliferation timescale helps propagate the tissue damage pattern.
(TIF)

**S3 Fig. 1D Parameter space with fixed $\gamma$.** Tissue recovery probability as a function of $\beta$, $\alpha$ with fixed $\gamma$ = 5 (hours) shows a low ratio of $\beta/\gamma^* = 0.47\pm0.07$ (red line) marking the transition above which recovery probability is greater than 0.95 (A). Variation in recovery probability keeping $\alpha/\gamma$ fixed at 0.15, 1.25, 5.25 (lines in blue, black, and magenta respectively) across values of $\beta/\gamma$ (B) and fixing $\beta/\gamma$ = 0.05, 0.25, 3.25 (lines in blue, black, magenta respectively) across values of $\alpha/\gamma$ (C).
(TIF)

**S4 Fig. Regions of maximum proliferation for different tissue sizes.** Average number of new hepatocytes (shown by the colorbars) created for fixed $\beta$ = 5 (hours) for $N$ = 16 (A,C) and $N$ = 20 (B, D). Lines indicate $\gamma/\beta^* = 3.22\pm0.59$ in (A) and $\gamma/\beta^* = 3.81\pm0.51$ in (B). (C,D) shows maximum proliferation as a function of $\gamma/\beta$ for fixed $\alpha/\beta$ = 0.25 (blue), 3.25 (black), 5.25 (magenta) for $N$ = 16 (C) and $N$ = 20 (D).
(TIF)

**S5 Fig. Gain in the healthy cell population.** Gain of healthy cells in simulations that ultimately die. Gain percentage is calculated as $(\langle max(H)\rangle - H_i)/H_i^* 100\%$. In general, very few healthy cells are gained in simulations that ultimately progress to complete cell death.
(TIF)

**S6 Fig. 2D visualization output at fixed $\beta$.** Healthy states are shown in green, stressed states in magenta and dead states in black. $\beta$ is fixed at 5(hours). Output trajectories for initial conditions with 4 stressed layers with $\alpha/\beta$ = 1 and $\gamma/\beta$ = 0.05 (A), 1.45 (B), 2.45 (C), 4.45 (D) and 9.05 (E).
(TIF)

**S7 Fig. 2D parameter space at fixed $\alpha$.** (A) Simulations on a hexagonal lobule with fixed $\alpha$ = 5 (hours). Coexisting states shown in red (marked with C), parameters for which the tissue always survives shown in yellow (marked with S) and regions where the tissue always dies shown in green (marked with D). Similar to the parameter space seen in Fig 8A, for ratios of $\beta/\alpha$ below 0.15 ($\alpha/\beta$>6.7), the transition from recovery to death goes through regions of bistable outcomes of both tissue survival and death. By increasing that ratio to around $\beta/\alpha$ = 0.25 ($\alpha/\beta$ = 4), a mixture of coexistence and survival states is seen which again transitions into regions characterized by death, recovery and coexistence as $\gamma/\alpha$ is increased. As $\beta/\alpha$ is further increased ($\alpha/\beta$ is decreased) pure coexisting states can be seen wedged between different recovery and death regions. (B) Average size of the number of healthy states. Visualization outputs along the dashed line shown in S8 Fig.
(TIF)

**S8 Fig. 2D visualization output at fixed $\alpha$.** Healthy states are shown in green, stressed states in magenta and dead states in black. $\alpha$ is fixed at 5 (hours). Output trajectories for initial conditions with 4 stressed layers for $\gamma/\alpha$ = 2 with $\beta/\alpha$ = 0.05 (A). Here a small $\beta$ results in converting healthy cells to stressed cells quite easily and the tissue dies. On increasing the ratio of $\beta/\alpha$ = 0.3, the tissue can show signs of survival (B). As the ratio is increased, tissue shows signs of coexistence $\beta/\alpha$ = 0.55 in (C) and $\beta/\alpha$ = 0.95 in (D) with different population levels of ($H$, $S$, $D$). For larger ratios, $\beta/\alpha$ = 1.15 in (E), the tissue shows signs of survival again.
(TIF)

**S9 Fig. Perturbations to the steady state populations.** Sample trajectories with the perturbation $S->H$ in (A-D) and $S->D$ (E-H) after the system has reached the steady coexistence state. Dotted lines indicates time of applied perturbation. Each panel consists of different trials. (A,B,E,F) are with $\alpha/\beta$ = 1, $\gamma/\beta$ = 2.45 and (C,D,G,H) are with $\alpha/\beta$ = 1, $\gamma/\beta$ = 4.45. All the values

in the left column are when 95% of stressed cells are converted into the healthy or dead states while the right column shows output when all but one stressed cell remains. Coexisting states are stable and the perturbations die down even after 95% removal of the stressed cells. (S1 Table).
(TIF)

**S10 Fig. Perturbation to the model parameters.** Initial parameters lead the system to a stable coexisting population after which the parameters are changed. (A-D) $\alpha/\beta = 1$, $\gamma/\beta = 2.45$. At Time=5000.25 (hours) marked by the red arrows, $\gamma/\beta$ is changed to 0.05 (A), 1.85 (B), 9.05 (C), 10.45 (D). (E-H) has $\alpha/\beta = 1$, $\gamma/\beta = 4.45$ where $\gamma/\beta$ is changed to 0.05 (E), 1.85 (F), 9.05 (G), 10.45 (H). System goes to the expected outcome at the new parameters.
(TIF)

**S11 Fig. Visualization output from the model parameter perturbations.** (A-B) has $\gamma/\beta = 2.45$ and a shift to 0.05 (A) and 9.05 (B) at Time=5000.25 (hours). (C-D) is with $\gamma/\beta = 4.45$ and a shift to 0.05 (C) and 9.05 (D) at Time=5000.25 (hours).
(TIF)

**S12 Fig. Baseline noise to the conversion rule.** Conversion rule with an additional small probability $\epsilon$ of making a healthy cell stressed even in the absence of other stressed cells for $\gamma/\beta = 2.45$ (A-C) and $\gamma/\beta = 4.45$ (D-F). Healthy cell fraction decreases with $\epsilon = 0.01$ (A,D), 0.05 (B, E), 0.1 (C,F) as compared to $\epsilon = 0$ in Fig 8E and 8F.
(TIF)

**S13 Fig. Asymmetry in the initial conditions.** (A) 2D initial conditions where cells in the four nearest layers next to the CV have 80% probability of being in a stressed state. (B) 2D state space with the modified initial conditions; $\beta = 5$ (hours); survival states are marked with S and colored in yellow, dead states are marked with D and colored in green and coexistence states are colored in red. (C) 2D state space with fixed $\alpha = 5$ (hours). Phase space outcomes are similar to Fig 8A.
(TIF)

**S14 Fig. Variation in the 2D lattice size and the initial number of damaged cells.** Simulations with a larger grid size with CV-PT length ($N_L$) of 50 ($N = 7650$) cells over 20 trials (A). Expansion of the coexistence region is seen. Different initial conditions with a fixed $N_L = 10$, $N = 330$ and number of stressed cell layers at 2 (B) and 8 (C) along the CV-PT axis. The SD boundary shifts to the left in (C) similar to the 1D results, but the coexistence region shows little changes.
(TIF)

**S15 Fig. Parameter spaces with fixed number of divisions.** Space with maximum number of allowed divisions at 1, (A) shows possible outcomes as a phase space (B) shows the average healthy population at the end of the simulation. (C-F) Increasing the allowed number of divisions per hepatocyte to 2 brings back the coexisting states. Parameters are for fixed $\beta = 5$ (hours) (C-D); fixed $\alpha = 5$ (hours) (E-F).
(TIF)

**S16 Fig. Fractional damage along the CV to PT axis at APAP dose of 250 mg/kg.** Fig shows the fractional damage along the CV to PT axis at 2, 6, 12 and 24 hours for an APAP dose of 250 mg/kg. Damage regions are relatively unchanged.
(TIF)

**S17 Fig. 3D simulations.** 3D simulations with 10 layers along a vertical z-axis, each consisting of hexagonal 2D lattice. Each cell in a hexagonal plane can access the neighboring site to the top and bottom planes, making the total number of nearest neighbors to 8. Simulations were done over 25 trials. Significant expansion of the coexisting regions are seen as compared to Fig 8A.
(TIF)

**S1 Table. Probabilities of tissue survival on applying different perturbations at steady state.**
(DOCX)

**S1 File.**
(ZIP)

## Acknowledgments

PA would like to thank John Metzcar for helpful discussions. Microscopy was conducted at the Biocomplexity Institute and at the Light Microscopy Imaging Center, both at Indiana University Bloomington.

## Author Contributions

**Conceptualization:** Priyom Adhyapok, James P. Sluka, Sherry G. Clendenon, James A. Glazier.

**Data curation:** Priyom Adhyapok, James P. Sluka, Sherry G. Clendenon, Victoria D. Sluka.

**Formal analysis:** Priyom Adhyapok, Xiao Fu, James P. Sluka, Sherry G. Clendenon.

**Funding acquisition:** James P. Sluka, Sherry G. Clendenon, Kenneth Dunn, James E. Klaunig, James A. Glazier.

**Investigation:** Priyom Adhyapok, Xiao Fu, Victoria D. Sluka, Zemin Wang.

**Methodology:** Priyom Adhyapok.

**Project administration:** James P. Sluka, Sherry G. Clendenon, James E. Klaunig, James A. Glazier.

**Resources:** Zemin Wang, Kenneth Dunn, James E. Klaunig.

**Software:** Priyom Adhyapok, Xiao Fu, James P. Sluka.

**Supervision:** James P. Sluka, Sherry G. Clendenon, James A. Glazier.

**Validation:** Xiao Fu, James P. Sluka, Sherry G. Clendenon.

**Visualization:** Priyom Adhyapok, Xiao Fu, James P. Sluka, Sherry G. Clendenon.

**Writing – original draft:** Priyom Adhyapok, James P. Sluka, Sherry G. Clendenon.

**Writing – review & editing:** Priyom Adhyapok, Xiao Fu, James P. Sluka, Sherry G. Clendenon, Kenneth Dunn, James E. Klaunig, James A. Glazier.

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
