## [Decision Letter · Decision Letter 0]

11 Sep 2020

PONE-D-20-21446

A computational model of liver tissue damage and repair

PLOS ONE

Dear Dr. Adhyapok,

Thank you for submitting your manuscript to PLOS ONE. The paper was sent to three reviewers, whose comments are appended below. I would be happy to reconsider the paper again after the manuscript is revised to address those comments.

We look forward to receiving your revised manuscript.

Kind regards,

Jordi Garcia-Ojalvo

Academic Editor

PLOS ONE

2. Our internal editors have looked over your manuscript and determined that it is within the scope of our Liver Diseases Call for Papers. This collection of papers is headed by a team of Guest Editors for PLOS ONE. Additional information can be found on our announcement page: https://collections.plos.org/s/liver-diseases. If you would like your manuscript to be considered for this collection, please let us know in your cover letter and we will ensure that your paper is treated as if you were responding to this call. If you would prefer to remove your manuscript from collection consideration, please specify this in the cover letter.

Reviewers' comments:

Reviewer's Responses to Questions

**Comments to the Author**

1. Is the manuscript technically sound, and do the data support the conclusions?

Reviewer #1: Yes

Reviewer #2: Yes

Reviewer #3: Yes

2. Has the statistical analysis been performed appropriately and rigorously? 

Reviewer #1: I Don't Know

Reviewer #2: No

Reviewer #3: N/A

3. Have the authors made all data underlying the findings in their manuscript fully available?

Reviewer #1: Yes

Reviewer #2: Yes

Reviewer #3: Yes

4. Is the manuscript presented in an intelligible fashion and written in standard English?

Reviewer #1: Yes

Reviewer #2: Yes

Reviewer #3: Yes

5. Review Comments to the Author

Reviewer #1: This manuscript is a solid piece of science which I encourage publication with a minor set of changes that I think will improve its impact and reach.

Line 94: Here is the first time pericentral and periportal are mentioned but, as scientist not well versed in anatomy, I would appreciate if you could briefly describe what it means the first time these terms are introduced.

115, 781: TNF-alpha

130: This interesting discussion about tipping scales and sudden transitions would be improved by some references about phase transitions from physics, pointing the reader to similar work (for instance percolation or infection dynamics)

142: The list of Celullar Automata references is quite short. It would be interesting to mention Forest Fire CA publications, as these recapitulate a conceptually similar process (3 states: empty, vegetation and fire, fire can propagate to neighboring vegetation and consumes itself creating an empty cell...) but also majority rule CAs and Ising model among others.

154: I would encourage the authors to move the mention that cells can indeed recover after being stressed here from the discussion were it currently is (line 782). The existence of such process indicates possible limitations of this model and needs to be justified as the model is explained.

181: It would be useful to state here directly that the automaton is synchronous. Do you think these results would be afected by using an asynchronous CA? There are families of solutions in lateral inhibition CAs that are artifacts caused by synchrony in the CA.

294: Bar = 300 microm

299, 590, 621: Different figures are given for necrosis to start, lower bounds of 1-2 hours and that it has begun by 4 h. However, if I understand correctly, the simulations use a beta probability that gives 5h average necrosis time, right? Given the interest to use experiments to inform the modeling, I would encourage the authors to consolidate these statements and offer an explanation as to why is 5h chosen.

432 and others: when there are phasepaces shown, could you describe how do you sample the space? how many simulations and is there any interpolation involved?

838: Any libraries or only custom code?

Additional notes: I think a very interesting aspect that is not explored very deeply in the manuscript is the spatial structure that can emerge even in stochastic CAs, especially in the parameter space where there is coexistence of cellular states. Correlation functions between sites (like with Ising model) or information content / structural analysis could be used to quantify patterning and structure. It would also provide a prediction to match with real data and further validate the model.

Reviewer #2: This manuscript by Adhyapok et. al presents a study drug injected APAP and how this can both lead to cell proliferation and cell death and tissue damage. In order to do this they construct a simple model, where cells can take 1 of 3 states, and introduce a rule where one cell can switch to another state dependent on specific rates. Using a 1-dimensional model, where every cell has two neighbours, they investigate parameter combinations, that induce the critical transitions from proliferating states to tissue damage states. They outline the ratio between key parameters that are important for this transition, and finds the critical threshold of damage pattern propagation.

I believe that the work on a complex system, using a simple model to gain insight in the most important cellular mechanisms can be of great value, however I feel that many of the results in general can be stated much more clearly and more rigorously.

General comments:

The manuscript could be rewritten and many sections shortened and made much sharper. I feel it is sometimes difficult to figure out where the authors are heading with entire sections. Also I would make the figures sharper and probably move a large number of the current figures into supplementary material in order to focus on the real essential figures. Furthermore I would add error bars on several figures, and for instance add a colorbar to every heat map.

Major revisions:

1) I don't think it is explained clearly why the 1D and 2D models are relevant for a problem that in reality is truly a 3D problem. I don't see why it should not be possible to carry out similar analysis in 3D and compare with the findings in lower dimensions.

2) Using rates, a cell will have an exponential waiting time to make a transition, however in reality there would probably be some time delay in this. The authors should address this, and show that it does not often occur that a specific cell changes state numerous times.

3) I find the results of coexistence states of great interest, but I would like to see how robust this result is to noise, for instance if there is a probability that a cell will be stressed even though the neighbourhood is not occupied by other stressed cells.

4) The majority of simulations are done with N being below 20. Why is it not realistic to consider N in one or two orders of magnitude larger? Would coexistence be more probable in this case?

5) Even though the comparison between experiments and modelling is nice, I fail to see convincing evidence that this model actually gives insight into the nature of liver damage. Does it produce testable predictions?

Reviewer #3: This contribution by Adhyapok et al. presents a computational model of hepatocyte behavior in the liver tissue after APAP induced liver damage. The manuscript models the progression of liver damage using a cellular automaton model and performs simulations of linear chains or grids of hepatocytes. Remarkably, model was informed with well-described experiments. This article carefully explores the parameter space for the three main parameters of the model and includes relevant predictions like a minimum healthy population size for tissue recovery or bistable states of tissue survival or death. Some comments are addressed to Authors for their consideration to improve this manuscript:

1. There are some typos and grammatical errors. Please check and correct them to improve the manuscript.

Line 129: Substitute ";" by "question;"

Line 126: Coma missing after "regeneration"

Lines 115, 294, 781: Missing symbols

Line 489: a space is missing before "an"

Line 493: a space is missing before "values"

Line 526: a space is missing "and"

Line 607: a space is missing before "is"

etc.

2. Please include the number of replicates used for each experiment (figures 2 and 3).

3. In the discussion section I missed an in-depth look at the impact of the model results on experimental observations. Please include some discussion on how these results would translate into an experimental context or the possible applications of the model results on the treatment of liver damage.

6. PLOS authors have the option to publish the peer review history of their article (what does this mean?). If published, this will include your full peer review and any attached files.

Reviewer #1: No

Reviewer #2: No

Reviewer #3: **Yes: **Victoria Doldán-Martelli

---

## [Author Response · Author response to Decision Letter 0]

2 Nov 2020

The authors would like to thank the reviewers for their efforts in reviewing this manuscript. Their comments have led to important improvements in the manuscript.

Reviewer 1:

Line 94: Here is the first time pericentral and periportal are mentioned but, as scientist not

well versed in anatomy, I would appreciate if you could briefly describe what it means the first

time these terms are introduced.

We thank the reviewer for this suggestion as this description was missing from the text. We have now added an introductory paragraph describing the anatomical terms used in the text.

115, 781: TNF-alpha

This has now been corrected in the text.

130: This interesting discussion about tipping scales and sudden transitions would be improved by some references about phase transitions from physics, pointing the reader to

similar work (for instance percolation or infection dynamics)

We thank the reviewer for pointing out these suggestions. We have now added references to phase transitions in infection dynamics and traffic flow.

142: The list of Celullar Automata references is quite short. It would be interesting to mention

Forest Fire CA publications, as these recapitulate a conceptually similar process (3 states:

empty, vegetation and fire, fire can propagate to neighboring vegetation and consumes itself

creating an empty cell...) but also majority rule CAs and Ising model among others.

We have now added additional references to the CA theory, including Forest Fire publications. We have also added references of majority rule CAs based on interacting spin systems used to study opinion dynamics. 

154: I would encourage the authors to move the mention that cells can indeed recover after

being stressed here from the discussion were it currently is (line 782). The existence of such

process indicates possible limitations of this model and needs to be justified as the model is

explained

We have now shifted the line from the discussion to follow the text describing the different transition rules. We believe that while this additional time-scale could be very relevant, it will not change the nature of the steady state solutions, but will only shift the boundaries of these domains in the phase plane. We have added this point to the discussion as well.

181: It would be useful to state here directly that the automaton is synchronous. Do you think

these results would be afected by using an asynchronous CA? There are families of solutions

in lateral inhibition CAs that are artifacts caused by synchrony in the CA.

While our system is updated at every time step (we have added this to the text) instead of permuting through the cell population one at a time and updating time after N changes have been made, the probabilistic nature of our updation rules helps in incorporating asymmetry to the cell state changes and randomizing the order in which updates are made. Additionally, proliferating cells are also shuffled if there are multiple contenders to fill the same empty site, eliminating other artifacts.

294: Bar = 300 microm

We have now corrected this in the text.

299, 590, 621: Different figures are given for necrosis to start, lower bounds of 1-2 hours and

that it has begun by 4 h. However, if I understand correctly, the simulations use a beta

probability that gives 5h average necrosis time, right? Given the interest to use experiments

to inform the modeling, I would encourage the authors to consolidate these statements and

offer an explanation as to why is 5h chosen.

Our main interest for the simulations is to make comments in terms of relative quantities of the timescales. For this purpose, the fixed timescale for each simulation set has been arbitrarily chosen with an interest to explore as much of the parameter space as finely as possible. Instead of simulations with quantities of say α=20 hours, we have relied on a fixed value for any parameter scan and at the end used the available APAP data to calibrate and compare the results for use in future mouse models. 

432 and others: when there are phasepaces shown, could you describe how do you sample

the space? how many simulations and is there any interpolation involved?

We have added information on the number of simulations used for the phase spaces. The varying timescales were incremented in steps of 0.50 (hours). We have also added this to the text. In the 1D case, as only two states were identified, the average healthy cell population as the only metric has been shown. In 2D and 3D, every simulation over all the trials was inspected - any parameter outcome that showed recovery, death or locked in coexistence were stored in separate lists. Then these lists were plotted in three different colors and the transparency adjusted to reflect regions in which multiple outcomes were possible. We have highlighted this in the text. No interpolation was used to generate the phase spaces, however best fit lines were used to describe the critical ratios in the 1D parameter space to find a value below which recovery probability is less than 0.95.

838: Any libraries or only custom code?

Custom codes were used for the simulations. The source code will be submitted.

Additional notes: I think a very interesting aspect that is not explored very deeply in the

manuscript is the spatial structure that can emerge even in stochastic CAs, especially in the

parameter space where there is coexistence of cellular states. Correlation functions between

sites (like with Ising model) or information content / structural analysis could be used to

quantify patterning and structure. It would also provide a prediction to match with real data

and further validate the model.

We thank the reviewer for this very interesting suggestion. We tried a few metrics to explore this - (1) Analysis of connected healthy cell clusters to see if there is any preferred size domain for any of the population fractions, (2) Cell-cell neighbor numbers to see if the number of healthy-stressed to healthy-healthy cells changed over time. We found no emergence of any spatial patterns of interest. More sophisticated techniques could be employed to probe this in greater detail as a separate question.

Reviewer 2:

The manuscript could be rewritten and many sections shortened and made much sharper. I

feel it is sometimes difficult to figure out where the authors are heading with entire sections.

Also I would make the figures sharper and probably move a large number of the current

figures into supplementary material in order to focus on the real essential figures.

Furthermore I would add error bars on several figures, and for instance add a colorbar to

every heat map.

Based on this suggestion to focus on the key results, we have now shortened a few sections and moved plots to the supplementary. 

 Results section 2: Plots with fixed γ have been moved to the supplementary and the heading of the section has been changed to focus on our key result.

 Results section 3: Heading has been changed to focus on our key result.

 Results section 4: Section has been shortened by removing several of the trajectory plots and the maximum gain plot has been moved to the supplementary section.

 Results section 6: This section has now been merged with the Discussion section to answer other points related to what the testable predictions for the model could be.

Each of the heat maps are now shown with their own color map.

1) I don't think it is explained clearly why the 1D and 2D models are relevant for a problem

that in reality is truly a 3D problem. I don't see why it should not be possible to carry out

similar analysis in 3D and compare with the findings in lower dimensions.

We thank the reviewer for this suggestion. 3D results for the same parameter range with a fixed β have now been added to the paper. In these simulations, a cell could have 6 neighbors on the same plane and two additional neighbors to the top and bottom to interact or divide into. Doing so significantly expanded the coexisting regions. Part of the rationale for including the 1D and 2D models is that these models have been known in literature to be able to account for many of the observations seen in experiments (for example, see refs. 1 and 2). Our results show qualitatively similar results in going from 2D to 3D simulations.

2) Using rates, a cell will have an exponential waiting time to make a transition, however in

reality there would probably be some time delay in this. The authors should address this, and

show that it does not often occur that a specific cell changes state numerous times.

We anticipate that it is indeed true that a healthy cell doesn’t have the proliferative capacity to keep dividing (stressed cells don’t change state multiple times anyway as they only progress towards necrosis in this model). To address this to an extent we have suggested the concept of an upper bound on the number of divisions, though other future methods can also address this in different ways.

3) I find the results of coexistence states of great interest, but I would like to see how robust

this result is to noise, for instance if there is a probability that a cell will be stressed even

though the neighbourhood is not occupied by other stressed cells.

The result of these simulations have now been added as a figure to the supplementary section. Adding a baseline probability of cells being stressed, in the absence of other neighbourhood stressed cells, reduced the healthy cell population fraction. Other step perturbations, where population levels are suddenly changed, have also already been discussed (S9 - S11 Figs).

4) The majority of simulations are done with N being below 20. Why is it not realistic to

consider N in one or two orders of magnitude larger? Would coexistence be more probable in

this case?

The values of N we used are based on the typical size of a lobule. Lobules ten times or more larger are not observed. The portal triad to central vein distance in a lobule is variable but only over a fairly small range. However, analysis of a much larger system could be interesting and we thank the reviewer for this suggestion. Larger scale simulations comprising N=7650 cells (N_L=50 along CV-PT axis) have now been added to the supplementary section; we indeed saw an expansion of the coexisting region for the same parameter sets. 

5) Even though the comparison between experiments and modelling is nice, I fail to see

convincing evidence that this model actually gives insight into the nature of liver damage.

Does it produce testable predictions?

We believe the result that rapid proliferation does not always ensure recovery of the tissue to be of great interest. Since ratios of our timescales define the parameter space of outcomes, future biological perturbations could check the predictions of critical ratios and the presence of minimum threshold of healthy cell population. In addition, the variability of outcomes in certain parameter domains suggest small inter-individual variations can result in different outcomes for APAP overdose. The highly variable response to APAP overdose in humans can perhaps be better understood by our results. There are of course multiple other explanations for inter-individual variation (such as changes in the balance between Phase I and Phase II metabolism) but our results suggest the variability can also arise downstream of the APAP metabolic events. For the mouse studies, we have suggested a few parameter ranges around which some of these predictions can be tested.

Reviewer 3:

Line 129: Substitute ";" by "question;"

Line 126: Coma missing after "regeneration"

Lines 115, 294, 781: Missing symbols

Line 489: a space is missing before "an"

Line 493: a space is missing before "values"

Line 526: a space is missing "and"

Line 607: a space is missing before "is"

etc

We thank the reviewer for pointing these mistakes out. We have now corrected them in the text.

2. Please include the number of replicates used for each experiment (figures 2 and 3).

We have now added the number of replicates used for each experiment to the text. 

3. In the discussion section I missed an in-depth look at the impact of the model results on

experimental observations. Please include some discussion on how these results would

translate into an experimental context or the possible applications of the model results on the

treatment of liver damage.

We have rewritten our discussion to focus on a few results that could lead to biological testing. Specifically, we found the result that rapid proliferation might not always be the answer to limiting liver damage to be of great interest. We have also moved our experimental comparison section to the discussion to provide suggestions on what the ranges of the three timescales for an APAP study could look like. Biological perturbations to these timescales could help test out our predictions and additionally indicate the presence of critical ratios of tissue recovery or the presence of a threshold of healthy cell population.

---

## [Decision Letter · Decision Letter 1]

23 Nov 2020

A computational model of liver tissue damage and repair

PONE-D-20-21446R1

Dear Dr. Adhyapok,

We’re pleased to inform you that your manuscript has been judged scientifically suitable for publication and will be formally accepted for publication once it meets all outstanding technical requirements.

Kind regards,

Jordi Garcia-Ojalvo

Academic Editor

PLOS ONE

Additional Editor Comments (optional):

Reviewers' comments:

Reviewer's Responses to Questions

**Comments to the Author**

1. If the authors have adequately addressed your comments raised in a previous round of review and you feel that this manuscript is now acceptable for publication, you may indicate that here to bypass the “Comments to the Author” section, enter your conflict of interest statement in the “Confidential to Editor” section, and submit your "Accept" recommendation.

Reviewer #1: All comments have been addressed

Reviewer #2: All comments have been addressed

2. Is the manuscript technically sound, and do the data support the conclusions?

Reviewer #1: Yes

Reviewer #2: Yes

3. Has the statistical analysis been performed appropriately and rigorously? 

Reviewer #1: Yes

Reviewer #2: Yes

4. Have the authors made all data underlying the findings in their manuscript fully available?

Reviewer #1: Yes

Reviewer #2: Yes

5. Is the manuscript presented in an intelligible fashion and written in standard English?

Reviewer #1: Yes

Reviewer #2: Yes

6. Review Comments to the Author

Reviewer #1: (No Response)

Reviewer #2: I think the questions have been seriously addressed and the text has in general been much improved compared to the first submission.

I therefore suggest that this paper is published.

7. PLOS authors have the option to publish the peer review history of their article (what does this mean?). If published, this will include your full peer review and any attached files.

Reviewer #1: No

Reviewer #2: **Yes: **Mathias Luidor Heltberg

---

## [Editor Report · Acceptance letter]

1 Dec 2020

PONE-D-20-21446R1 

A computational model of liver tissue damage and repair 

Dear Dr. Adhyapok:

I'm pleased to inform you that your manuscript has been deemed suitable for publication in PLOS ONE. Congratulations! Your manuscript is now with our production department. 

Kind regards, 

on behalf of

Dr. Jordi Garcia-Ojalvo 

Academic Editor

PLOS ONE